

# Numerical Framework for the Computation of Urban Flux Footprints Employing Large-eddy Simulation and Lagrangian Stochastic Modeling

Mikko Auvinen[1,2], Leena Järvi[1], Antti Hellsten[2], Üllar Rannik[1], and Timo Vesala[1]

[1]Department of Physics, P.O. Box 64, University of Helsinki, 00014 Helsinki, Finland
[2]Finnish Meteorological Institute, P.O. Box 503, 00101 Helsinki, Finland

*Correspondence to:* Mikko Auvinen (mikko.auvinen@helsinki.fi)

**Abstract.** Conventional footprint models cannot account for the heterogeneity of the urban landscape imposing a pronounced uncertainty on the spatial interpretation of eddy-covariance (EC) flux measurements in urban studies. This work introduces a computational methodology that enables the generation of detailed footprints in arbitrarily complex urban flux measurements sites. The methodology is based on conducting high-resolution large-eddy simulation (LES) and Lagrangian stochastic (LS) particle analysis on a model that features a detailed topographic description of a real urban environment. The approach utilizes an arbitrarily sized target volume set around the sensor in the LES domain, to collect a dataset of LS particles which are seeded from the potential source-area of the measurement and captured at the sensor site. The urban footprint is generated from this dataset through a piecewise post-processing procedure, which divides the footprint evaluation into multiple independent processes that each yield an intermediate result that are ultimately selectively combined to produce the final footprint. The strategy reduces the computational cost of the LES-LS simulation and incorporates techniques to account for the complications that arise when the EC sensor is mounted on a building instead of a conventional flux tower. The presented computational framework also introduces a result assessment strategy which utilizes the obtained urban footprint together with a detailed land cover type dataset to estimate the potential error that may arise if analytically derived footprint models were employed instead. The methodology is demonstrated with a case study that concentrates on generating the footprint for a building-mounted EC measurement station in downtown Helsinki, Finland, under neutrally stratified atmospheric boundary layer.

## 1 Introduction

Micrometeorological measurements in densely built city environments pose an antipodal problem: They are essential in establishing the fundamental basis for the study of urban microclimate, but these measurements are endowed with pronounced uncertainties, which mainly originate from the topographic and elemental complexity of the urban landscape. The resulting noncompliance between the theory and practice in urban micrometeorological measurements undermines the study on how our cities interact with the surrounding atmosphere. At the very heart of this discord lies the problem concerning the determination of effective source-areas, or footprints, of urban flux or concentration measurements.



The footprint is a concept used to describe the surface area that contains the sources and sinks which contribute to the measured quantity obtained by a sensor (Pasquill, 1972). In another words, it is such sensor's 'field of view' whose identification is essential in interpreting the obtained flux or concentration values in their correct spatial extent (Schmid, 2002). Mathematically, the footprint is a transfer function $f$, which relates the value of a measurement $\eta$ at location $\boldsymbol{x}_{\mathrm{M}} = (x_{\mathrm{M}}, y_{\mathrm{M}}, z_{\mathrm{M}},)$ to the spatial

distribution of flux or concentration sources $Q$ from a volumetric domain $\Omega$ of interest:

$$\eta(\boldsymbol{x}_{\mathrm{M}}) = \int\limits_{\Omega} f(\boldsymbol{x}_{\mathrm{M}}, \boldsymbol{x}') Q(\boldsymbol{x}') \, d\boldsymbol{x}'. \tag{1}$$

Thus, the footprint can also be interpreted as a spatial weighting function that expresses the probability with which a fluid element that coincides with an element of $Q$ contributes to the measurement at $\boldsymbol{x}_{\mathrm{M}}$ (Pasquill and Smith, 1983). In accordance with Sogachev et al. (2005), this study does not adhere to the strict interpretation where the footprint is only a function of

turbulent diffusion and source-sensor location, but allows the possibility that, for instance, variations in source-area topography can influence the result. Consequently, the footprint should provide the critical link between the point measurement and the geographical distribution of sources, yielding a complete characterization of $\eta$ with regard to its contents. In effort to achieve this, analytical closed-form solutions have been derived for the footprint functions – see Schmid (2002) for a comprehensive review – but only under the assumptions that (1) steady-state conditions prevail during the analyzed period, (2) turbulent

fluctuations in the atmospheric boundary layer (ABL) are horizontally homogeneous, and (3) there is no vertical advection. These assumptions allow the governing equations to be reduced to a time-averaged balance between advection and turbulent diffusion which admits, with appropriate parametrization of turbulent flow field, a closed-form expression for the footprint function.

The underlying assumptions are often acceptable in measurement sites where the sensors are mounted on towers that have

been appropriately placed above homogeneous forested landscapes and well above the surface roughness sublayer height where the effects of the individual roughness elements disappear. However, due to practical regulations constraining measurement campaigns in densely populated cities, sufficiently tall flux towers cannot be erected above the skyline of central urban areas. It is often inevitable that if the urban microclimate is to be studied experimentally, the measurements must be extracted near the border of the roughness sublayer by sensors that are mounted either on low-rise towers or on top of tall buildings. In

these suboptimal conditions, assumption (2) becomes strictly invalid and assumption (3) highly questionable because urban boundary layer (UBL) flows are typically characterized by developing and strongly heterogeneous flow conditions particularly at lower elevations where individual buildings influence the turbulence.

Considering that the analytical footprint models effectively provide ellipse-shaped probability distributions for the source contributions without any regard to topographic heterogeneities, it becomes clear that the use of such source-area models

becomes highly suspect in real urban conditions. This is an unacceptable state of affairs in the urban micrometeorology research and immediately calls for targeted efforts to alleviate the uncertainties associated with the invaluable urban flux-measurement data. Although, the first efforts by Vesala et al. (2008), utilizing the method by Sogachev et al. (2002), already explored topography-sensitive urban footprints, the applicability of the documented approach has not reached the scale and accuracy requirement of the urban footprint problems considered herein.





As a response, this works introduces a new numerical methodology to construct detailed topography-sensitive footprints for complex urban flux measurement sites by the means of pre- and post-processing developments and a large-eddy simulation (LES) solver suite that features an embedded Lagrangian stochastic (LS) particle model. This coupled model will be referred to with an acronym LES-LS. The proposed methodology is designed to be first and foremost a post-processing procedure, which

exploits the current state-of-the-art LES-LS modeling framework in an urban setting with a minimal investment in the initial setup.

The principal objective is to provide a reliable computational framework, founded on a high-resolution LES-LS analysis, to generate the most accurate footprint estimates feasible without the need to conduct tracer gas experiments, which are nearly impossible to arrange in residential areas. These computationally generated footprints open up the possibility to study the

appropriate placement of new measurement stations and to assess the magnitude of the potential misinterpretation which may arise from the application of closed-form footprint models to urban flux or concentration measurements. The proposed framework is also supplemented by a convenient technique to approximate this error with the assistance of a land cover classification dataset.

The methodology is demonstrated with a numerical case study, which is staged in Helsinki, the coastal capital city of Finland,

and focuses on the eddy-covariance (EC) measurement site mounted on the roof of Hotel Torni (Nordbo et al., 2015; Kurppa et al., 2015), which is the tallest accessible building in the downtown region. The EC sensor is situated 2.3 m above the building, corresponding 74 m above the sea level, whereas the building height is 57.7 m . Thus, the measurement height is $z_\mathrm{M} = 60$ m above the surrounding ground level while the mean building height of the surrounding area is 24 m. The sensor is judged to be situated at the edge or just above the roughness sublayer (Nordbo et al., 2013). The site belongs to SMEAR III (Station

for Measuring Ecosystem-Atmosphere Relations, Järvi et al., 2009) and is also part of the urban network of atmospheric measurement sites (Wood et al., 2013). Its potential source-area closely resembles a typical European city arrangement that features perimeter blocks with inner courtyards.

This study employs the PArallelised LES Model PALM (Maronga et al., 2015; Raasch and Schröter, 2001), which has been previously applied to footprint studies by Steinfeld et al. (2008) and very recently by Hellsten et al. (2015) who constructed

footprints for an idealized city environment as a precursor study to this work. The presented contribution places special emphasis on the issue of composing footprints for flux measurement sites that are surrounded by arbitrarily heterogeneous topography and may be compromised by the fact that they are mounted on top of actual buildings instead of conventional radio-mast-like towers. Such complex urban setting requires a new mechanism for constructing footprints, which is accompanied by a requirement that the associated LES-LS simulation is capable of resolving the relevant turbulent structures ranging from the street

canyon scale phenomena within the roughness sublayer to the larger ABL structures, while also accounting for the interaction between them (Anderson, 2016).



## 2 Materials and methods

### 2.1 Numerical modeling framework

The PALM model utilized in this study is an open source numerical solver for atmospheric and oceanic flow simulations. The software has been carefully designed to run efficiently on massively parallel supercomputer architectures and it is therefore

exceptionally well-suited for high-resolution UBL simulations considered herein. The LES model employs finite-difference discretization on staggered Cartesian grid and utilizes an explicit Runge-Kutta time-stepping scheme to solve the evolution of velocity vector $\boldsymbol{u} = (u, v, w)$, modified perturbation pressure $\pi^*$, potential temperature $\theta$ and specific humidity $q_v$ fields from the conservation equations for momentum, mass, energy and moisture respectively. The conservation equations are implemented in an incompressible, Boussinesq-approximated, non-hydrostatic and spatially filtered form, which indicates that

the conservation of mass is imposed by the solution to a Poisson equation for $\pi^*$. The filtering refers to the separation of scales in LES where the turbulent scales containing the majority of energy are resolved by the grid while the diffusive effect of the unresolved subgrid-scale (SGS) turbulence is accounted for by a SGS turbulence model. To achieve closure in the final system of equations, PALM implements the 1.5-order SGS turbulence model by Deardorff (1980), modified according to Moeng and Wyngaard (1988) and Saiki et al. (2000). The model involves an additional prognostic equation for SGS turbulent kinetic

energy (SGS-TKE) $e$.

The embedded Lagrangian particle model in PALM implements the time-accurate evolution of discrete particles (either with or without mass) through a technique that conforms to the LES approach: the trajectories are integrated in time such that the transporting velocity field is decomposed into deterministic (i.e. resolved) and stochastic (i.e. subgrid-scale) contributions. The deterministic velocity components are directly obtained from the LES solution, while the random components are evaluated

according to Weil et al. (2004). Although LS modeling approaches that are less computationally expensive exist (Glazunov et al., 2016), warranting further investigation on their applicability to urban problems, the presented high-resolution urban flow problem is assumed to require the highest level of description also from the LS model; the interaction between the atmospheric wind and the cascade of multistoried buildings and street canyons gives rise to strongly anisotropic turbulence structures, which are not reliably amendable to parametrization.

While the LES-LS simulations are carried out in large supercomputing facilities, the pre-processing of the urban topography model and the post-processing of the final footprint from raw data is performed on a personal workstation utilizing freely available numerical scripting and data visualization technologies. See Section 5 for availability.

### 2.2 Urban LES setup and analysis

#### 2.2.1 Urban topography model

The urban topography model used in describing the bottom wall boundary of the LES domain, is prepared from a detailed 2 m resolution laser-scanned dataset of the Helsinki area (Nordbo et al., 2015). The data is conveniently available in raster map

**Figure 1.** Raster maps of topography height $h$ (left) and land cover types $LC$ (right) from Helsinki area. The rectangle in the bottom left corner is aligned with south-westerly wind and represents the area of interest for the footprint analysis. In the surface type classification each pixel (2 m) is categorized according to the following numbering: 0=building, 1=impervious (rock, paved, gravel), 2=grass, 3=low vegetation, 4=high vegetation and 5=water.

format and, in addition to the height distribution $h(x,y)$, also includes land cover type classification $LC(x,y)$ which are both shown in Fig. 1. Access to similar surface data source is a critical pre-requisite for the presented methodology.

The horizontal domain for the LES analysis extends $L_x = 4096$ m in the mean wind direction and $L_y = 2048$ m in the crosswind direction and is spatially oriented such that $x$-axis is coincident with the geostrophic wind direction of the case study. The EC measurement site at Hotel Torni is pivotally located in the LES domain to facilitate the determination of its footprint. However, the extracted raster map has to be first purposefully pre-processed to attain a form that complies with the LES analysis-specific requirements. The following manipulations were applied to obtain the final topography model depicted in Fig. 2:





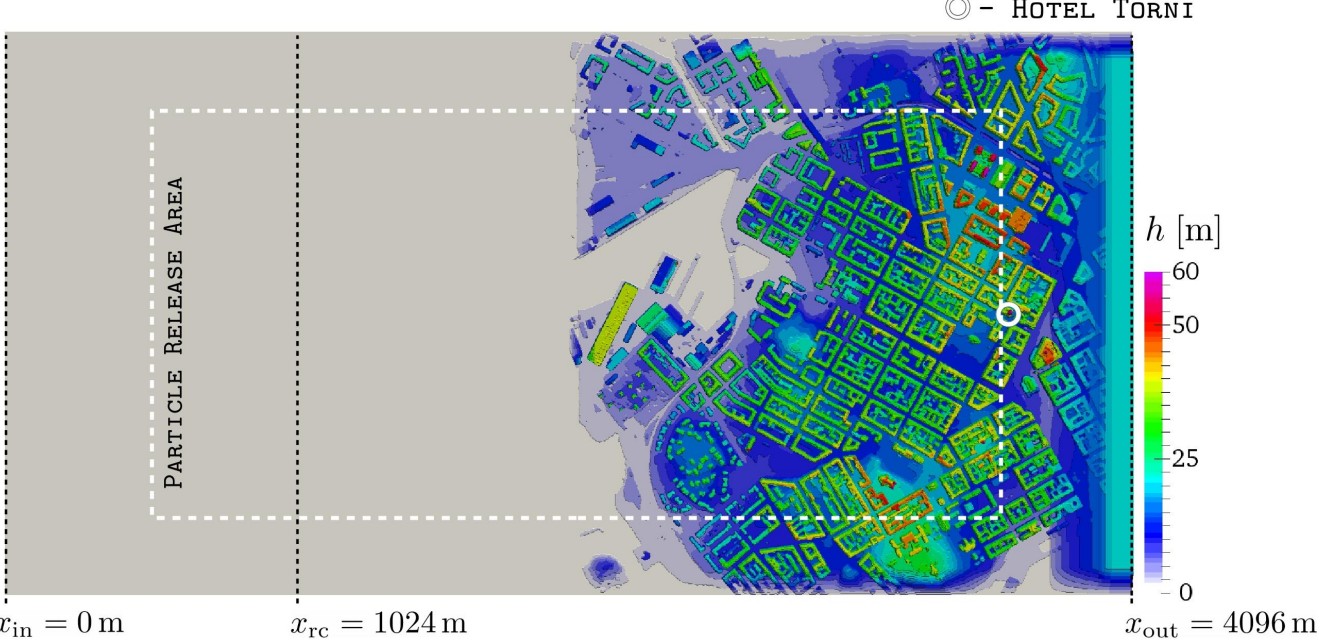

**Figure 2.** Visualization of the topography height distribution underlying the LES domain. The particle release area is enveloped by a white dashed line. The location of the turbulence recycling plane is marked by a black dotted line at $x_{\mathrm{rc}}$.

1. The first half of the topography model (where $x < L_x/2$) is flattened for the purpose of generating physically realistic ABL conditions at the inlet through turbulence recycling technique (see below).

2. The lateral sides were made identical for cyclic boundary condition treatment by applying a zero-height margin that smoothly blends toward the values in the interior.

3. Immediately upstream of the outlet boundary, a margin with sloping terrain height is applied to force the highly turbulent flow (caused by the buildings near the end of the domain) to slightly accelerate before reaching the outlet boundary where reversed flow causes numerical difficulties.

### 2.2.2 Physical setup for the LES model

The meteorological conditions for the simulation are adopted from September 9$^{\mathrm{th}}$ in 2012 when near neutral ABL conditions were recorded with the EC measurements made in Torni. Lidar measurements (Wood et al., 2013) from the chosen time frame yielded $|\boldsymbol{u}_g| = 10\,\mathrm{m\,s^{-1}}$ for the geostrophic wind in south-westerly direction ($\alpha = 218°$) and $\delta \approx 300$ m for the boundary layer height. The Coriolis force (corresponding to latitude 60°N) is included to account for the turning of the flow within the boundary layer. The meteorological conditions are conveyed to the simulation by the means of a pre-computed ABL solution over flat surface, which in this context represents the surface of the Baltic Sea bordering Helsinki from the south. The boundary





conditions for the velocity solution in this *precursor* simulation were set such that a slip wall condition is applied at the top and a no-slip condition at the bottom boundary of the domain while setting all the lateral boundaries as periodic.

For the precursor simulation the solver was run with an option that explicitly conserves the initial mass flow rate across the system which was specified by initializing the velocity field by a constant value $\boldsymbol{u}\big|_{t=0} = 0.95\boldsymbol{u}_g$. This initialization value was determined by trial and error with the objective that the precursor solution would ultimately yield the desired geostrophic wind value at $z = \delta$ for the temporally and horizontally averaged velocity field $\langle \bar{\boldsymbol{u}} \rangle^{\mathrm{pre}}$. The boundary layer growth was controlled by initializing the potential temperature field with a vertical profile $\theta_0(z)$ that features a strong inversion layer at $300 < z < 350$ m. This $\theta_0(z)$-profile is defined according to the following lapse rates:

$$
\frac{\partial \theta_0}{\partial z} = \begin{cases} 0\ \mathrm{K\,km^{-1}} & 0\,\mathrm{m} \leq z < 300\,\mathrm{m} \\ 50\ \mathrm{K\,km^{-1}} & 300\,\mathrm{m} \leq z < 350\,\mathrm{m} \\ 3\ \mathrm{K\,km^{-1}} & 350\,\mathrm{m} \leq z \end{cases}
$$

The precursor LES solution was computed on a grid that has the same resolution and vertical dimension as the principal urban LES grid, but its lateral dimensions are smaller by an integer division. Table 1 summarizes the respective grid characteristics. The study features a spatial resolution of 1 m, which is unprecedented at this scale. The same resolution was found sufficient by Giometto et al. (2016) to capture the relevant turbulence physics within a real urban roughness sublayer. However, the effect of grid resolution on the final result is not investigated in this work.

The precursor simulation generates a highly resolved ABL solution that will be utilized, first, in a recursive manner to initialize the entire urban LES flow field with turbulence and, second, to aid constructing appropriate inlet boundary conditions though a technique labeled *turbulence recycling*, which is based on the method by Lund et al. (1998) with modifications by Kataoka and Mizuno (2002). The implementation of this boundary condition in PALM is presented in Maronga et al. (2015), but to aid discussion the description is also covered here with modified notation.

Denoting prognostic field variables by $\psi = \psi(\boldsymbol{x}, t)$ where $\psi \in \{u, v, w, \theta, e\}$, the precursor solution is used to extract temporally and horizontally averaged vertical profiles $\langle \bar{\psi} \rangle^{\mathrm{pre}}(z)$ for the turbulence recycling boundary condition. These stationary profiles are utilized at the inlet boundary in the urban simulation to conserve the global state of the mean flow, but in a manner that also incorporates physically sound turbulent fluctuations that occur in an ABL flow. This is achieved by specifying a recycling plane, that is, an $yz-$plane at a windwise coordinate $x_{\mathrm{rc}}$, placed sufficiently far downstream from the inlet to prevent feedback of disturbances between the two planes. The fluctuations are obtained from the recycling plane through the following technique

$$
\psi'\big|_{x=x_{\mathrm{rc}}} = \psi\big|_{x=x_{\mathrm{rc}}} - \langle\psi\rangle_y\big|_{x=x_{\mathrm{rc}}}
$$

where the spatial mean (in the crosswind direction) $\langle\psi\rangle_y = \langle\psi\rangle_y(z, t)$ at the recycling plane is computed as a time dependent vertical profile

$$
\langle\psi\rangle_y\big|_{x=x_{\mathrm{rc}}} = \frac{1}{N_y} \sum_{i=1}^{N_y} \psi(x_{\mathrm{rc}}, y_i, z, t)
$$




**Table 1.** Computational grid specifications.

|  | Resolution $\Delta x, \Delta y, \Delta z$ | Dimensions $N_x \times N_y \times N_z$ | Total no. of grid points $\approx N_{tot}$ |
|---|---|---|---|
| precursor grid | $1\,\text{m}, 1\,\text{m}, 1\,\text{m}$ | $1024 \times 512 \times 512$ | $\approx 67 \times 10^6$ |
| urban grid | $1\,\text{m}, 1\,\text{m}, 1\,\text{m}$ | $4096 \times 2048 \times 512$ | $\approx 4295 \times 10^6$ |

that carries a dependence on $N_y$. Finally, utilizing the precursor generated mean profiles, the turbulence recycling inlet boundary condition becomes

$$\psi\big|_{x=x_{\text{in}}} = \big\langle \bar{\psi} \big\rangle^{\text{pre}} + \psi'\big|_{x=x_{\text{rc}}} . \tag{2}$$

In this study, the recycling plane is situated, as shown in Fig. 2, in accordance with the precursor domain length such that $(x_{\text{rc}} - x_{\text{in}}) = 1024\,\text{m} \approx 3.4\delta$ and the same distance is allocated from the recycling plane to the edge of the urban topography to ensure that disturbances originating from the urban terrain are not conveyed back to the inlet. The chosen turbulent inlet arrangement generated no observable feedback effect on the incoming turbulence field.

### 2.2.3  LS Particle model Setup for the footprint evaluation

The embedded LS particle model is employed such that, after the initial transients in the LES solution have subdued (after approximately 5 min of simulation), the release of particles is activated within the region outlined in Fig. 2. The release area extends 3030 m ($\approx 41 z_{\text{M}}$) in the upwind direction and 780 m ($\approx 10.5 z_{\text{M}}$) in both lateral directions from the Hotel Torni's EC site. The release area has been trimmed according to preliminary trial simulations to reduce the number of redundant particles in the domain.

Denoting the Lagrangian coordinate vector of the $l^{\text{th}}$ particle by $\boldsymbol{X}^l(t) = \big(X^l(t), Y^l(t), Z^l(t)\big)$, the release locations $\boldsymbol{X}_o^l = \boldsymbol{X}^l\big|_{t=0}$ are uniformly distributed 2 m apart in the $x$- and $y$-directions while the vertical coordinate is set $\Delta Z_o = 1\,\text{m}$ above the topography: $Z_o^l = h(X_o^l, Y_o^l) + \Delta Z_o$. The release height of one grid spacing at 1 m resolution is inferred to be a justifiably close to the surface to represent both the traffic emissions as well as the surface atmosphere exchanges. It also lowers the risk of accumulating a large number of particles within the first grid cell where the velocity values are dictated by the logarithmic wall function and the vertical advection of particles solely by the stochastic model due to Weil et al. (2004). Thus, the underlying assumption is that, at 1 m resolution, the release height of 1 m does not influence significantly the footprint distribution, which is evaluated at 2 m horizontal resolution.

The raw particle data for constructing footprints through LES-LS modeling in an arbitrarily heterogeneous environment is obtained by setting a target volume around the specified sensor location $\boldsymbol{x}_{\text{M}}$ and recording, which particles hit this target. Although this approach appears natural and straight-forward at first sight, a closer scrutiny reveals a number of problematic issues which arise with this setup, particularly when the flux sensor is mounted on a building (or close to one) instead of a tower. Purely from the perspective of particle data acquisition in the LES-LS simulation, setting a larger target volume would directly





alleviate the computational effort required to gather a large enough dataset of particle hits, but this would clearly violate the formal premise that the footprint should be evaluated for the coordinate $x_M$ of the sensor. However, it turns out that the discrete setting of the LES-LS approach questions the relevance of seeking an urban footprint for a precise point near the surface of a solid structure.

5  Consider the problem of fixating on the exact location $x_M$ of the sensor. This effort becomes immediately futile as the spatial resolution with which the buildings are described in the topography model (which contains information on elevation changes only) cannot account for structural details that, in reality, influence the flow conditions at the precise location of the sensor. The same reasoning also extends to the LES flow analysis where the computational cost would become prohibitively expensive if the resolution would be set according to the $\sim 10^{-1}\,\mathrm{m}$ scale of structural detail of building facades and roof-tops

10 in the hypothetical situation that such datasets were available. Therefore, it is important that the methodology for evaluating footprints in urban environments comes with a pre-requisite that the resolution demands of the LES-LS model are purely dictated by the turbulent structures within the urban canopy and not the fine details of the sensor site. On these grounds, the method to collect particle data in the LES-LS simulation is based on setting a finite target volume around the sensor location $x_M$ without strictly dictating the appropriate size. This is done understanding the fact that the flow around the sensor mounting

15 building strongly interacts with the flow resulting in strong gradients in the mean velocity field in the vicinity of the sensor. This is bound to further complicate the subsequent post-processing of the flux footprint because the eddy-covariance approach necessitates that the effect of the mean flow should be eliminated through the process of coordinate rotation (Aubinet et al., 2012), which is presented in the context of this study in Section 2.3.1. Clearly, the discrete LES-LS approach in an arbitrarily complex urban environment is endowed with pronounced uncertainties. For this reason, the post-processing procedure has

20 to encompass a capability to conduct spatial sensitivity analysis on the intermediate footprint results and, according to its outcome, selectively exploit the particle dataset in the final processing of the result.

  Adopting this strategy reduces the level of rigor required at the setup stage of the LES-LS analysis and simplifies the guidelines for the particle acquisition: The target volume should be centered at $x_M$ and its dimensions chosen to represent the sensor site proportionately (vagueness intended) to the dimensions of the building geometry. In all cases, it is important to

25 acknowledge that, as a rule of thumb, more than $10^7$ particle hits need to be recorded at the target volume during the course of the LES-LS simulation to gather a large enough dataset for flexible post-processing. In general, it pays off to specify an oversized target and gather a large dataset accepting that it contains certain percentage of particle hits whose contribution will be discarded. In this study the target for monitoring particle hits is specified as a box of volume $\mathcal{V}_T = \Delta x_T \times \Delta y_T \times \Delta z_T = 8\,\mathrm{m} \times 20\,\mathrm{m} \times 12\,\mathrm{m}$. The box is centered at the apex of Hotel Torni (which closely coincides with the actual sensor location $x_M$)

30 such that it extends 10 m in the crosswind and upward directions, 4 m in both streamwise directions and 2 m downward entering partly into the building structure. And, to reiterate, these dimensions were chosen under the guiding principle that the target box reasonably represents the sensor site and enables particle hits to be gathered at higher rate. Fig. 3 provides an illustration of the size and placement of the target box in relation to the surrounding urban topography. The monitoring is performed at $0.5$ s intervals, which corresponds to ca. 8 LES time steps. This allows the same particle to be recorded multiple times at different

35 locations within the target box. This feature is intentional and desirable because of the chosen post-processing strategy.



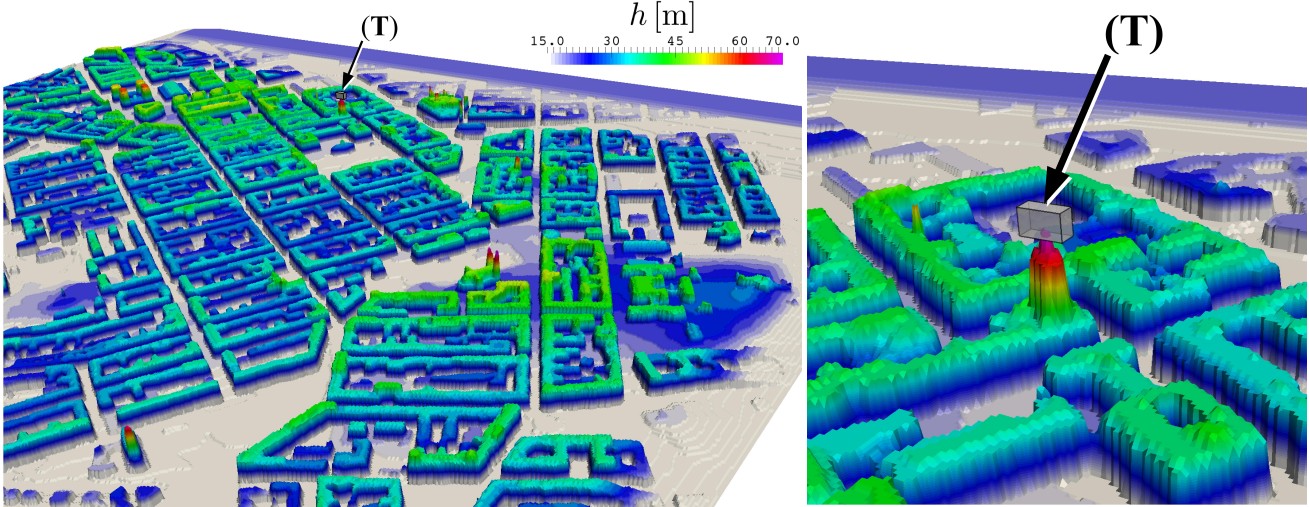

**Figure 3.** A three-dimensional rendering of the urban topography near Hotel Torni (left) and a close-up featuring the target box (T) for particle capturing (right).

### 2.2.4 LES-LS analysis

The precursor simulation is run for 1 h physical time to develop the ABL turbulence sufficiently and the temporal averaging is activated for the latter 45 min. The initialization of the primary LES-LS computation with this precursor solution expectedly results in short-lived unphysical fluctuations around the urban topography, but after 3 minutes of simulation these overshoots

have been advected away from the domain. The release of LS particles is initiated after 5 minutes of simulation and from there on particles are released simultaneously in puffs at 10 s intervals such that two particles are seeded from each location at every instance. This translates into releasing approximately $2.36 \times 10^6$ particles every interval. The release schedule was determined by trial and error to best utilize the computational capacity of the supercomputer. Each particle is assigned a maximum lifetime $T^l_{max} = 1200$ s, which is long enough to guarantee that even the particles that are advected by the slowest $\sim 0.2\boldsymbol{u}_g$ velocity

scales, manage to travel over 2 km during this time frame. The total number of particles in the whole domain converged to approximately $68 \times 10^6$. Particles reaching any of the lateral boundaries or the top boundary are 'absorbed', that is, deleted and deallocated from the computer's memory while the wall boundary below functions as an ideally smooth reflective surface for the particles. The simulation was run for 3 h physical time during which ca. $19 \times 10^6$ particle hits were recorded at the target volume. The computation cost of this simulation is comparable to running 3-4 urban flow simulations with the objective

on studying turbulence. The LS model constituted merely 20% of the total cpu-time of the LES-LS simulation, which is an appreciably moderate value considering the high number of particles handled by the solver.



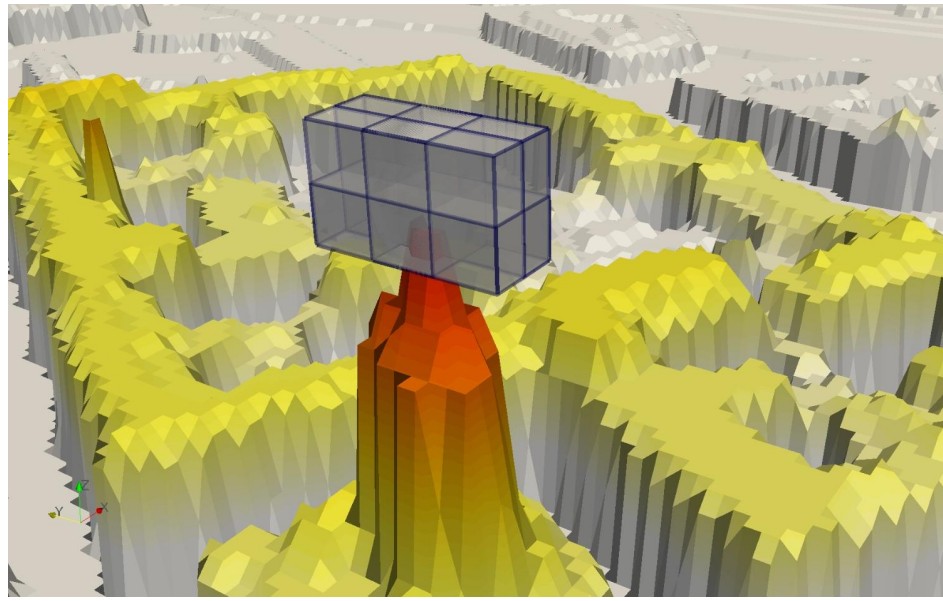

**Figure 4.** Example discretization of target volume $\mathcal{V}_{\mathrm{T}}$ into $n_x \times n_y \times n_z$ subvolumes. A coarse illustration with $n_x = 2$, $n_y = 3$, and $n_z = 2$ is shown.

## 2.3 Piecewise post-processing methodology for constructing the footprint

During the LES-LS simulation, the sampling of particle hits at the target volume $\mathcal{V}_{\mathrm{T}}$ entailed recording each $l^{\mathrm{th}}$ particle's coordinate of origin $\boldsymbol{X}_o^l$, incident velocity $\boldsymbol{U}_{\mathrm{T}}^l = \left(U_{\mathrm{T}}^l, V_{\mathrm{T}}^l, W_{\mathrm{T}}^l\right)$ at the target, and the associated sample location $\boldsymbol{X}_{\mathrm{T}}^l$ (indicating where the particle hit the target) ultimately giving rise to a large dataset

$$\mathtt{S} = \left\{ (\boldsymbol{X}_o, \boldsymbol{U}_{\mathrm{T}}, \boldsymbol{X}_{\mathrm{T}})^l \mid l \in \{1, \dots, N_{rp}\}, \left(\boldsymbol{x}_{\mathrm{M}} - \frac{\Delta \boldsymbol{x}_{\mathrm{T}}}{2}\right) \le \boldsymbol{X}_{\mathrm{T}}^l \le \left(\boldsymbol{x}_{\mathrm{M}} + \frac{\Delta \boldsymbol{x}_{\mathrm{T}}}{2}\right) \right\} \tag{3}$$

where $N_{rp}$ refers to the total number of released particles.

According to the issues discussed in Section 2.2.3, the post-processing of $\mathtt{S}$ is now required to account for the spatial uncertainty and facilitate a sensitivity study on the obtained result. This is achieved by introducing a piecewise processing strategy where the principle idea is that the original dataset $\mathtt{S}$ is split into smaller subsets according to a Cartesian discretization of the target volume $\mathcal{V}_{\mathrm{T}}$. See an example illustration in Fig. 4. Thus, the target is divided into subvolumes $\mathcal{V}_{i,j,k}$, satisfying $\mathcal{V}_{\mathrm{T}} = \sum_k^{n_x} \sum_j^{n_y} \sum_i^{n_z} \mathcal{V}_{i,j,k}$ where the $i, j, k$ are the Cartesian indices of the subvolumes. The number of divisions in each coordinate direction $n_x$, $n_y$, and $n_z$ have to be determined case by case as the optimal values depend on the target volume size, the total number of particle entries in the dataset and the complexity of flow solution in the vicinity of $\boldsymbol{x}_{\mathrm{M}}$.

Each target subvolume now yields an associated subset $\mathtt{s}_{i,j,k} \subset \mathtt{S}$ containing a record of the particles that hit the corresponding subvolume $\mathcal{V}_{i,j,k}$ centered at $\boldsymbol{x}_{\mathcal{V}_{i,j,k}} = \boldsymbol{x}_{\mathrm{M}} + d\boldsymbol{x}_{i,j,k}$, where $d\boldsymbol{x}_{i,j,k}$ is the displacement from the exact measurement location $\boldsymbol{x}_{\mathrm{M}}$ to the center of the subvolume $\mathcal{V}_{i,j,k}$. The obtained subsets can be independently post-processed to generate sectional flux footprints $f_{i,j,k}$ utilizing an estimator similar to Kurbanmuradov et al. (1999) (see also Rannik et al., 2000), but





modified to approximate the footprint by computing the probability with which a fluid parcel released from a continuous source at $\boldsymbol{x}_f = (x, y, h + \Delta Z_o)$ will lie within $\mathcal{V}_{i,j,k}$ at any given time. Discretizing the source-area (i.e. footprint grid) by $\Delta \boldsymbol{x}_f = (\Delta x_f, \Delta y_f, 0)$, the estimator reads

$$f_{i,j,k}(\boldsymbol{x}_f) = \frac{1}{N_{i,j,k} \Delta x_f \Delta y_f} \sum_{l}^{N_{i,j,k}} \frac{W_{\mathrm{T}}'^l}{|W_{\mathrm{T}}'^l|} \, I \tag{4}$$

which has an implicit dependence on the vicinity of $\boldsymbol{x}_{\mathcal{V}_{i,j,k}}$ through the spatial confinement of $\mathbf{s}_{i,j,k}$. In Eq. (4) $N_{i,j,k}$ denotes the number of particles entries within the subset $\mathbf{s}_{i,j,k}$ collected over a sufficiently long time period, and

$$W_{\mathrm{T}}'^l = \left( W_{\mathrm{T}}^l - \langle \bar{w} \rangle_{i,j,k} \right) \tag{5}$$

is the vertical velocity deviation of the $l^{\mathrm{th}}$ particle from the spatially averaged mean flow value evaluated over the subvolume $\mathcal{V}_{i,j,k}$. Equation (5) relates to the coordinate rotation of the EC sensor, which eliminates the effect of $\bar{w}$ from the vertical flux evaluation by aligning the sensor with the mean wind (Aubinet et al., 2012, p.76). Here, the evaluation of $W_{\mathrm{T}}'^l$ proves particularly problematic due to the approximations associated with the use of $\langle \bar{w} \rangle_{i,j,k}$ and, therefore, it is a subject of further discussion in Section 2.3.1. Finally, the function $I = I(\boldsymbol{X}_o^l, \boldsymbol{x}_f, \Delta \boldsymbol{x}_f)$, which is responsible for distributing the hits on to the footprint grid based on the particles' coordinate of origin, is given as follows

$$I(\boldsymbol{X}_o^l, \boldsymbol{x}_f, \Delta \boldsymbol{x}_f) = \begin{cases} 1 & \text{if} \quad \boldsymbol{x}_f - \frac{\Delta \boldsymbol{x}_f}{2} \leq \boldsymbol{X}_o^l < \boldsymbol{x} + \frac{\Delta \boldsymbol{x}_f}{2} \\ 0 & \text{elsewhere} \end{cases} \tag{6}$$

The evaluation procedure (4) closely resembles that of Rannik et al. (2003), with the exception that here it is assumed that each particle is represented only once in each subset $\mathbf{s}_{i,j,k}$.

By default, the individual sectional footprints are evaluated from subsets that typically contain $\sim 10^5$ particle data entries, which is not a sufficient number to obtain a converged solution for the weight distribution. This is anticipated and acceptable in the piecewise approach because the objective is to inspect and compare the individual distributions which should, however, reveal the characteristic shape of the distribution readily. For this reason, each $f_{i,j,k}$ should be individually stored as a standalone two-dimensional scalar field (i.e. raster map) that can be projected onto the three-dimensional topography model of the LES domain to permit descriptive visualizations in the urban setting. The value of the denominator $D_{i,j,k} = N_{i,j,k} \Delta x_f \Delta y_f$ featuring in Eq. (4) has to be stored together with the footprint distribution because the assembly of the final footprint is carried out by computing

$$f = \frac{\sum_{i,j,k \in \mathsf{K}} D_{i,j,k} f_{i,j,k}}{\sum_{i,j,k \in \mathsf{K}} D_{i,j,k}} \tag{7}$$

where $\mathsf{K}$ is the set of all $i, j, k$ combinations which have been selected via spatial sensitivity analysis (covered in Section 2.3.2).

### 2.3.1 Coordinate rotation via farfield correction

The piecewise processing of the footprint carries an inherent difficulty that arises in situations where the mean flow displays strong gradients within the target volume. This is evidently present in the considered case study featuring an EC sensor mounted

close to top of a building. The difficulty relates to the evaluation of $\langle \bar{w} \rangle_{i,j,k}$ which is used in the footprint evaluation to extract the fluctuating velocity components about the mean value within its corresponding subvolume (as shown above). The initially attempted approach naturally involved utilizing LES to obtain the (45 min time-averaged) mean velocity distribution $\bar{w}$ from within the target box volume and evaluating the spatial average $\langle \bar{w} \rangle_{i,j,k}$ for each subvolume $\mathcal{V}_{i,j,k}$. However, it became

evident that this approach gave rise to a systematic negative bias in all footprints. This outcome persisted despite refining the discretization of the target volume (by increasing $n_x$, $n_y$, and $n_z$) which, however, is constrained by the number of particle data entries in each $\mathbf{s}_{i,j,k}$.

The issue is best demonstrated by an example case setting where the target volume is split into $n_x \times n_y \times n_z = 80$ subvolumes by specifying $n_x = 4$, $n_y = 5$, and $n_z = 4$. This resulted in subsets that contained approximately $2 \times 10^5$ data entries, indicating

that further target volume refinement is not advised because the quality of the sectional footprints would be too severely compromized. All the sectional footprints – and thus also any accumulated combination of them – came out such that the global integrals $\bar{f}^{\Omega} = \int_{\Omega} f \, d\boldsymbol{x}$ were negative mainly because the distributions featured consistently negative far fields. This is clearly exhibited in Fig. 5, which depicts a sign distribution over the topography and a crosswind integrated plot of a footprint, which has been obtained by the standard coordinate rotation in the piecewise processing approach.

It is recognized that the footprints in complex urban applications are not required to be positive definite as negative patches are expected to appear in the presence of individual obstacles or in larger scale due to changing vertical particle concentration profiles along the mean wind direction, which can cause more of the particles from the far field to approach the sensor from above (Horst and Weil, 1992; Finnigan, 2004; Steinfeld et al., 2008). Such net effect can also be brought about the vertical deflection in the mean flow caused by the internal boundary layer over the urban topography observed in the simulation. How-

ever, the behavior of $\bar{f}^y$ in Fig. 5 brings this into doubt: The negative far field of the footprint gradually extends well beyond the urban landscape while sustaining a negative slope (thus strengthening the negative contributions with distance) until reaching an asymptotic level at a substantial negative value. Under the assumption that the magnitude of the gradient $\left| \left( \frac{\partial \bar{f}^y}{\partial x} \right) \right|$ will not increase beyond the farthest upstream distance from the sensor (2760 m), no physical justification will be provided herein for such a dominant negative far field. Instead, in this context the cause of this behavior is concluded to be a consequence of the

numerical approach employed in the coordinate rotation. The far field asymptote value is highly sensitive to the coordinate rotation, which hinges upon the evaluation of $\langle \bar{w} \rangle_{i,j,k}$ whose applicability in the presence of high mean velocity gradients carries approximations. In the presence of high mean velocity gradients, utilizing the spatial average of the mean over $\mathcal{V}_{i,j,k}$ is associated with an assumption that the particle hit locations $\boldsymbol{X}_{\mathrm{T}}$ within $\mathcal{V}_{i,j,k}$ are uniformly distributed and the gradient of the mean vertical velocity field $\nabla \bar{w}$ and the ensemble average of the incident particle velocities $\nabla \langle W_{\mathrm{T}}^{\prime l} \rangle$ can be well approximated

by the same constant vector (i.e. the planes defining the local mean profiles would be aligned). Under such conditions, the balance of negative and positive contributions would be unaffected by the formulation. However, the evidence shows that this assumption fails as the footprints obtain a notable bias. An alternative approach, which would carry the potential to remedy this, would involve determining mean vertical velocity value $\bar{w}(\boldsymbol{X}_{\mathrm{T}}^l)$ *per particle* through linear interpolation utilizing the nearest available values stored on the LES grid. Unfortunately, this strategy is associated with implementation difficulties in the

piecewise post-processing approach, consequently motivating the development of a new technique labeled *far field correction*





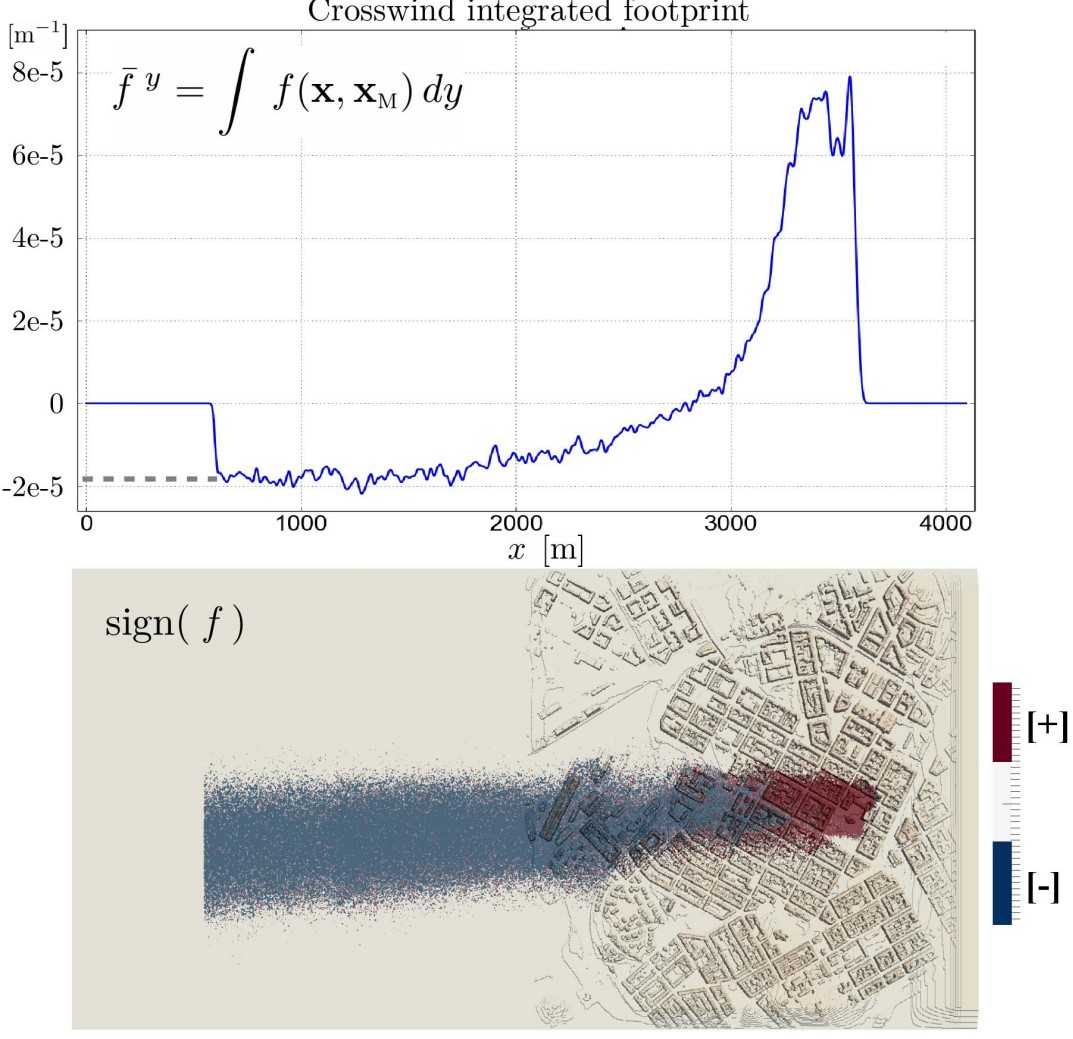

**Figure 5.** Crosswind integration (above) and sign (below) distributions of a footprint obtained using $\langle \bar{w} \rangle_{i,j,k}$ from LES solution in the piecewise post-processing procedure.

which incorporates well into the proposed post-processing strategy. The method fundamentally relies on the following simple assertion: If the footprint distribution plateaus in the far field, this asymptote can be declared as the zero reference level, which deviates from the 'correct' asymptote by a negligibly small offset.

Accepting this assertion and the associated approximation paves the way for a corrective coordinate rotation scheme which

5    can be laid out by first classifying the data contributing to the far field footprint via subsets $\mathbf{r}_{i,j,k} \subset \mathbf{s}_{i,j,k}$ which are defined as the sets of particle entries whose $X_o$ fall into the outermost portion of the domain

$$\mathbf{r}_{i,j,k} = \left\{ \mathbf{s}_{i,j,k} \,\middle|\, 0 \le \left( X_o^l - X_o^{min} \right) < \frac{\beta}{100} \left( x_{\mathrm{M}} - X_o^{min} \right) \right\}.$$



**Table 2.** Diagnostic data from the application of far field correction in the coordinate rotation. The farthest 15% of the source-area in the LES domain is considered (i.e. $\beta = 15$).

|  | Target Volume Discretizations: $(n_x \times n_y \times n_z)$ | | | Units |
|---|---|---|---|---|
|  | $(2 \times 3 \times 2)$ | $(3 \times 5 \times 3)$ | $(4 \times 5 \times 4)$ |  |
| mean$(c_{i,j,k})$ | 0.85 | 0.89 | 0.90 |  |
| std$(c_{i,j,k})$ | 0.09 | 0.11 | 0.15 |  |
| mean$(\langle \bar{w} \rangle_{i,j,k})$ | 1.16 | 1.13 | 1.15 | $\mathrm{m\,s^{-1}}$ |
| std$(\langle \bar{w} \rangle_{i,j,k})$ | 0.32 | 0.40 | 0.41 |  |

Here $X_o^{min} = \min\limits_{l \in \{1,\dots,N_{rp}\}} (X_o^l)$ is the farthest upstream coordinate where particles are seeded (thus, farthest away from $\boldsymbol{x}_{\mathrm{M}}$) and $\beta$ specifies the remotest percentage of the footprint across which the mean value of $\bar{f}^y$ no longer changes, that is, $\left\langle \frac{\partial \bar{f}^y}{\partial x} \right\rangle \approx 0$ when averaging over the length of the far field. (The $\beta$-value is case specific, but a typical range is expected fall between 10 and 20.) With the help of the far field datasets $\mathbf{r}_{i,j,k}$, the fluctuating vertical velocity component, used in Eq. (4) and previously defined by Eq. (5), can now be evaluated as

$$W_{\mathrm{T}}'^l = \left( W_{\mathrm{T}}^l - \langle \bar{w} \rangle_{i,j,k}^* \right) \tag{8}$$

where

$$\langle \bar{w} \rangle_{i,j,k}^* = c_{i,j,k} \langle \bar{w} \rangle_{i,j,k} \tag{9}$$

defines the far field corrected mean vertical velocity, which is obtained by scaling the initially obtained value by a coefficient $c_{i,j,k}$ to satisfy the criteria that the particle entries in each $\mathbf{r}_{i,j,k}$ do not contribute to the corresponding $f_{i,j,k}$. This becomes a simple one dimensional optimization problem in which the objective is to minimize $J = \left| \int_{\Omega_\beta} f_{i,j,k} d\boldsymbol{x} \right|$, where $\Omega_\beta$ represents the far field domain, by the means of controlling $c_{i,j,k}$. Because the control variable is a single scalar, a rudimentary implementation of an iterative gradient decent search algorithm suffices (see, for instance, Nocedal and Wright (2006)).

Table 2 displays selected diagnostic data obtained from an application of this far field correction technique to the Hotel Torni footprint case study. The data indicates that, when the mean vertical velocity values are initially obtained from the LES solution, the $c_{i,j,k}$ scaling coefficients concentrate near the mean value of 0.9. The range of individual values naturally depends on the magnitude of the starting value $\langle \bar{w} \rangle_{i,j,k}$ in Eq. (9) which, in turn, depend on the chosen discretization of the target volume. But, it is important to emphasize that, although the far field correction method is guaranteed to yield a physically justifiable asymptotic behavior for the footprint, the combined effect of the correction method and the target volume discretization on the final footprint result cannot be inferred from Table 2. For this purpose, the final assembly of the footprint must be accompanied by a sensitivity study.





### 2.3.2 Selective assembly of the final footprint

Since its conception it has been clear that the piecewise post-processing approach must be endowed with the capacity to incorporate a sensitivity analysis phase into the final assembly of the footprint result. One of the driving motivators for developing the piecewise approach arose from the need to reduce the computational cost of collecting a large number of particle hits by

a small target volume around $\boldsymbol{x}_\mathrm{M}$. However, the reduction can only be achieved by the piecewise post-processing approach if the sectional footprint results are allowed to be handled in an inadequately converged state. This is an important stipulation without which the proposed post-processing strategy fails to offer considerable computational savings.

Thus, the process of selectively assembling the final footprint result begins by first defining an inadequately converged initial footprint, which represents the desired preform at $\boldsymbol{x}_\mathrm{M}$. This reference footprint, labelled $f^\mathrm{REF}$, should be constructed from at

least $10^6$ particle entries to facilitate a sufficiently informative evaluation of sensitivities. The selection process proceeds by iteratively introducing partial contributions $f^{(l)}$ that are independent from $f^\mathrm{REF}$ and evaluating the sensitivity of the footprint distribution with respect to the selection of target box indices in $\mathtt{K}$ (see Eq. (7)). The objective is to obtain a sufficiently converged footprint while minimizing the discrepancy between the constituent $f_{i,j,k}$ included in the final result. Thus, the selection process is quantitatively guided by the evaluation of 'deltas' between $f^\mathrm{REF}$ and $f^{(l)}$, constructed from a partial set $\mathtt{K}^{(l)}$

of target box indices, (i.e. $\Delta f^{(l)} = f^\mathrm{REF} - f^{(l)}$) and utilizing a norm over a subdomain $\Omega_\star \subset \Omega$ encompassing only the near field (ca. 30% of the total length of the LES footprint domain) as a measure for the associated discrepancy. The near field norm is computed as

$$||\Delta f^{(l)}||_{2,\Omega_\star} = \left( \int\limits_{\Omega_\star} \left| \Delta f^{(l)}(\boldsymbol{x}_f) \right|^2 d\boldsymbol{x}_f \right)^{1/2} \tag{10}$$

utilizing identically normalized footprints for this evaluation. In this study the footprints are normalized to yield $\int_\Omega f d\boldsymbol{x} = 1$.

The exclusion of the outer portion of the footprint domain allows the relevant deviations in the near field to be reflected in $||\Delta f^{(l)}||_{2,\Omega_\star}$ while avoiding the contamination due to poorly defined 'deltas' in the weakly converged outer region. The search for the fitting contributions entails an iterative procedure, which is described herein for the case study utilizing target volume discretization $n_x \times n_y \times n_z = 3 \times 5 \times 3$. The relevant intermediate results and $||\Delta f^{(l)}||_{2,\Omega_\star}$ values are depicted in Fig. 6.

The process begins by setting at the $0^\mathrm{th}$ iteration $\mathtt{K}^{(0)} = \{\mathtt{I}_\mathrm{M}, \mathtt{J}_\mathrm{M}, \mathtt{K}_\mathrm{M}\} \subset \mathtt{K}$ where the indices correspond to the subvolume

containing $\boldsymbol{x}_\mathrm{M}$. The obtained footprint $f^{(0)} = f_{\mathtt{I}_\mathrm{M},\mathtt{J}_\mathrm{M},\mathtt{K}_\mathrm{M}}$, shown in Fig. 6, is composed of ca. $4 \times 10^5$ particle entries which is not sufficient to act as $f^\mathrm{REF}$. Thus, through a qualitative inspection, the original set is augmented $\mathtt{K}^{(1)} = \mathtt{K}^{(0)} + \{\mathtt{I}_\mathrm{M}, \mathtt{J}_\mathrm{M} \pm 1, \mathtt{K}_\mathrm{M}\}$ to yield $f^{(1)}$, which is chosen as the reference footprint.

The iterative process continues such that new candidate contributions $f^{(l)}$ are introduced incrementally in a radially outward progressing manner. This process is demonstrated in Fig. 6 where intermediate entries $f^{(2)}$-$f^{(6)}$ introduce differently

combined additions in $y$-, $x$- and $z$-directions. For the sake of brevity, the example contributions combine a relativel large number of $f_{i,j,k}$ entries. The decision to include a candidate contribution in the final assembly is done according to a criteria $||\Delta f^{(l)}||_{2,\Omega_\star} \le ||\Delta f||_\mathrm{max}$, where the maximum allowable discrepancy $||\Delta f||_\mathrm{max}$ must be determined according to the case-specific requirements. In this case study, the threshold was set to include $f^{(3)}$ in Fig. 6 such that $||\Delta f||_\mathrm{max} = ||\Delta f^{(3)}||_{2,\Omega_\star}$.





**Figure 6.** Illustration of the selective assembly of the final footprint for $n_x \times n_y \times n_z = 3 \times 5 \times 3 = 45$. Values of $||\Delta f^{(l)}||_{2,\Omega_\star}$ indicating discrepancy between $f^{\mathrm{REF}}$ and $f^{(l)}$ are shown where applicable. Acceptable candidates are marked by ✓ and the rejected by ✗. Note the use of short-hand notation, e.g. $1:3 = 1, 2, 3$.

The obtained final result, which combines the earlier accepted additions, features 20/45 of all subvolume contributions. Subsequently, the lowest vertical ($k = 1$) plane and the farthest ($i = 3$) plane were completely excluded from K in the final assembly. The obtained footprint exhibits adequate convergence also in the far field having been constructed from ca. $8 \times 10^6$ particle entries.



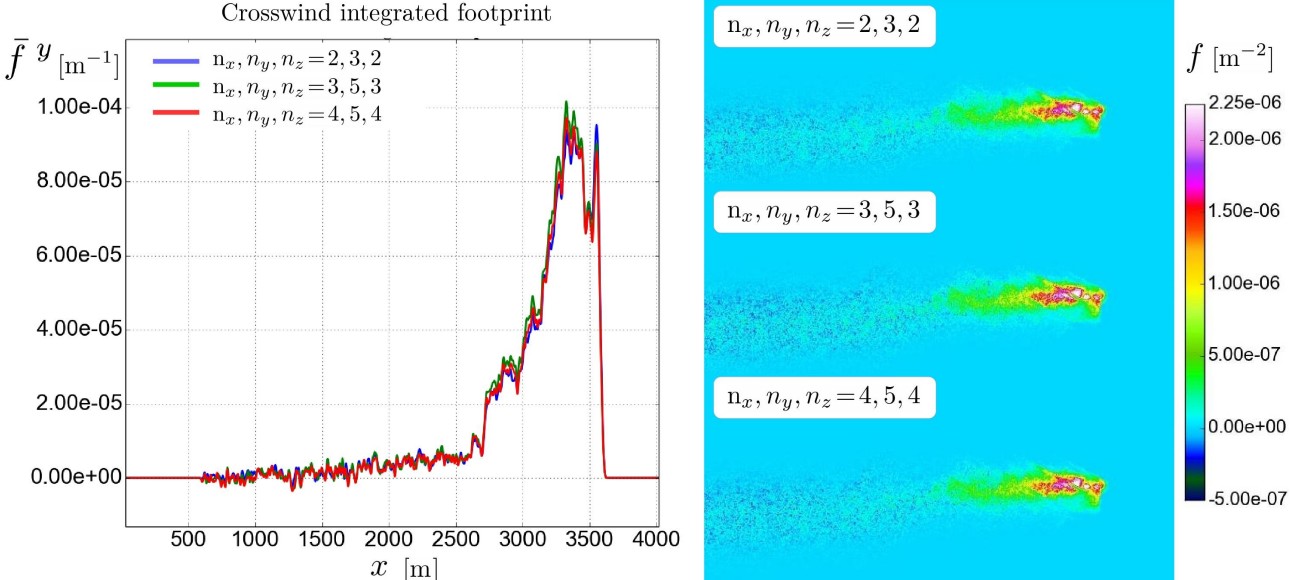

**Figure 7.** The final footprint results with different target volume discretizations.

As long as the individual subsets contain a sufficient number of particle data entries, it is beneficial to discretize the target volume as finely as possible (by increasing $n_x$, $n_y$, and $n_z$) as it enables a more flexible and fine-tuned assembly and permits a more accurate coordinate rotation treatment. When the post-processing techniques are implemented with appropriate automations, the labor cost is not significantly affected by the total number of subvolumes. A juxtaposition of three final results obtained through an identically guided selection process, but utilizing different combinations of $n_x$, $n_y$, and $n_z$ is depicted in Fig. 7. The comparison reveals that the differences between the three results are remarkably insignificant, which also indicates that the obtained footprint is not highly sensitive to the sensor placement. This demonstrates the utility and robustness of the selective piecewise post-processing approach. From here on the presented results correspond to the $n_x = 3$, $n_y = 5$, $n_z = 3$ target volume discretization level.

### 2.3.3 Outline of the procedure

Taking into account the far field correction procedure, the post-processing procedure for evaluating a footprint from a LES-LS obtained dataset can be described in the following steps:

1. Split the original dataset S into $n_x \times n_y \times n_z$ number of subsets labeled $\mathtt{s}_{i,j,k}$ according to a Cartesian division of the target volume $\mathcal{V}_\mathrm{T}$.

2. Evaluate an approximate footprint in a piecewise manner by applying Eq. (4) for each subset $\mathtt{s}_{i,j,k}$ and assemble the result according to Eq. (7) by selecting all $i, j, k$ values. (Here is it possible to use inaccurate data for the evaluation of $\langle \bar{w} \rangle_{i,j,k}$ as the objective is only to identify the far field)





3. Inspect the approximate footprint result to identify the extent of the far field (by specifying $\beta$) where the footprint reaches an asymptotic level to a good approximation and specify $\beta$ for the purpose of constructing $\mathbf{r}_{i,j,k}$

4. Evaluate the sectional footprints $f_{i,j,k}$ from corresponding $\mathbf{s}_{i,j,k}$ subsets by applying Eq. (4) with $\langle \bar{w} \rangle^*_{i,j,k}$ evaluated through far field correction approach as follows:

   (a) Select initial guess for $c^o_{i,j,k}$ and $\langle \bar{w} \rangle^o_{i,j,k}$, and utilizing the data from $\mathbf{r}_{i,j,k}$ compute the initial sectional footprint $f_{i,j,k} = f_{i,j,k}(c^o_{i,j,k})$ and the corresponding far field integral $J^o = \left| \int_{\Omega_\beta} f_{i,j,k}(c^o_{i,j,k}) d\boldsymbol{x} \right|$

   (b) Perturb the coefficient $c_{i,j,k} = c^o_{i,j,k} + dc$ (initially with a guessed perturbation $dc$) and, using $\langle \bar{w} \rangle^*_{i,j,k} = c_{i,j,k} \langle \bar{w} \rangle^o_{i,j,k}$ and the data from $\mathbf{r}_{i,j,k}$, compute $f_{i,j,k} = f_{i,j,k}(c_{i,j,k})$ and $J = \left| \int_{\Omega_\beta} f_{i,j,k}(c_{i,j,k}) d\boldsymbol{x} \right|$

   (c) Exit the loop if $J < \varepsilon$, where $\varepsilon$ specifies the tolerance

   (d) Compute derivative $\frac{dJ}{dc} = \frac{(J - J^o)}{dc}$ and specify a new perturbation from $dc = -\gamma \frac{dJ}{dc}$, where $\gamma > 0$ is a scaling parameter which, in this context, has been a experimentally set to ensure that the minimization problems converge sufficently

   (e) Set $J^o = J$, $c^o_{i,j,k} = c_{i,j,k}$ and return to step 4b

5. Select the appropriate set $\mathtt{K}$ of $i, j, k$ combinations employing sensitivity analysis procedure in Section 2.3.2

6. Assemble the final footprint via Eq. (7)

It is noteworthy that in step 2 for the approximate footprint evaluation and in step 4a for the initialization of the optimization loop, the values for the mean vertical velocities $\langle \bar{w} \rangle_{i,j,k}$ do not have to be accurate. Therefore, the use of vertical velocity data from LES can be omitted altogether, which simplifies the case setup and data handling considerably. The approximate values can be obtained more simply, for instance, by evaluating the mean of incident vertical velocity value from particle data in each $\mathbf{s}_{i,j,k}$.

## 3    Result assessment

The proposed methodology that is founded on high resolution LES-LS analysis and a piecewise post-processing approach has been shown to be a reliable, robust and accessible, although computationally expensive, approach to generate topography-sensitive footprints in real urban applications. Since the underlying motivation for this development effort sprung from the need to evaluate the potential error that may arise when analytical, closed-form footprint models are applied to urban flux measurements, this work also proposes a technique to approximate the magnitude of this error in the absence of field validation studies. This approach hinges on the assumption that, in a real urban application, a topography-sensitive footprint obtained through a highly resolved LES-LS analysis features a higher level of accuracy and a lower level of uncertainty than any available closed-form footprint model.

The proposed assessment technique compares the obtained LES-LS footprint result to an analytical model, which would otherwise be employed in similar studies, by applying the footprint distributions to the land cover classification $LC$ dataset in





**Table 3.** Parameters used in the Korman and Meixner footprint model.

| KM model parameter | Value | Explanation |
|---|---|---|
| Measurement height | 60 m | Hotel Torni measurement height |
| Mean wind speed | $4.86\,\mathrm{m\,s^{-1}}$ | EC measurement |
| Standard deviation of $v$ | $0.75\,\mathrm{m\,s^{-1}}$ | EC measurement |
| Roughness length ($z_0$) | 2.4 m | 10% of avg. building height (24 m) |
| Obukhov length | 10000 m | Neutrally stratified boundary layer, EC measurement |

Fig. 1 that is presented in the same resolution as the topography height. In the following demonstration the closed-form footprint model by Korman and Meixner (2001) (KM), which is widely utilized in the EC community (e.g., Christen et al., 2011; Kotthaus and Grimmond, 2012; Nordbo et al., 2013), is used as an example analytical model. This choice is subjective and implies no preference over other available footprint models (e.g., Kljun et al., 2015; Horst, 2001). The KM model parameters and their specific values are declared in Table 3. The mean wind speed and the standard deviation of the crosswind component are extracted from Hotel Torni's anemometer measurements gathered on September 9$^{\text{th}}$ 2012 during the same 30 min time frame that was used to specify the meteorological conditions for the LES simulation (see Section 2.2.2).

A preliminary comparison between the obtained LES-LS and KM footprint distributions, $f_{\text{LES}}$ and $f_{\text{KM}}$ respectively, in the considered Hotel Torni case study draws immediate attention to the apparent differences that become discernible from the juxtaposition displayed in Fig. 8. The shown distributions have been normalized to yield $\int_{\Omega} f\,d\boldsymbol{x}' = 1$ to aid the comparison. The LES-LS generated footprint exhibits complex, unpredictable probability distribution and a more pronounced spatial confinement, lacking the gradual asymptotic behavior of analytical models. Particularly the crosswind diffusion of the system is clearly over-predicted by the KM model. The most evident deviations occur in the near field, where the $f_{\text{LES}}$ exhibits strong local variations between building tops and street canyons. Moreover, examining the crosswind integrated footprints in Fig. 9 reveals how $\bar{f}_{\text{LES}}^{y}$ reacts abruptly to changes in the example urban landscape, leveling off to a shallow decending slope much earlier than the gradually declining curve of $\bar{f}_{\text{KM}}^{y}$. Thus, the presented comparison in the context of this case study succeeds in laying bare the nontrivial nature of urban footprints and highlights the importance of utilizing a high-resolution LES-LS approach to examine complex urban EC measurement sites.

### 3.1 Virtual assessment technique

The comparative technique proposed for assessing the potential error that may arise if urban measurements are interpreted with closed-form footprint models, exploits the land cover dataset under the assumption that the $LC$ distribution conveys the inherent urban heterogeneity sufficiently. Under this premise, the $LC$ distribution can be adopted as a model distribution of sources $Q$ such that each $e^{\text{th}}$ land cover type is assigned a constant mean source strength $\langle Q_e \rangle = \text{const}$. Thus, under this





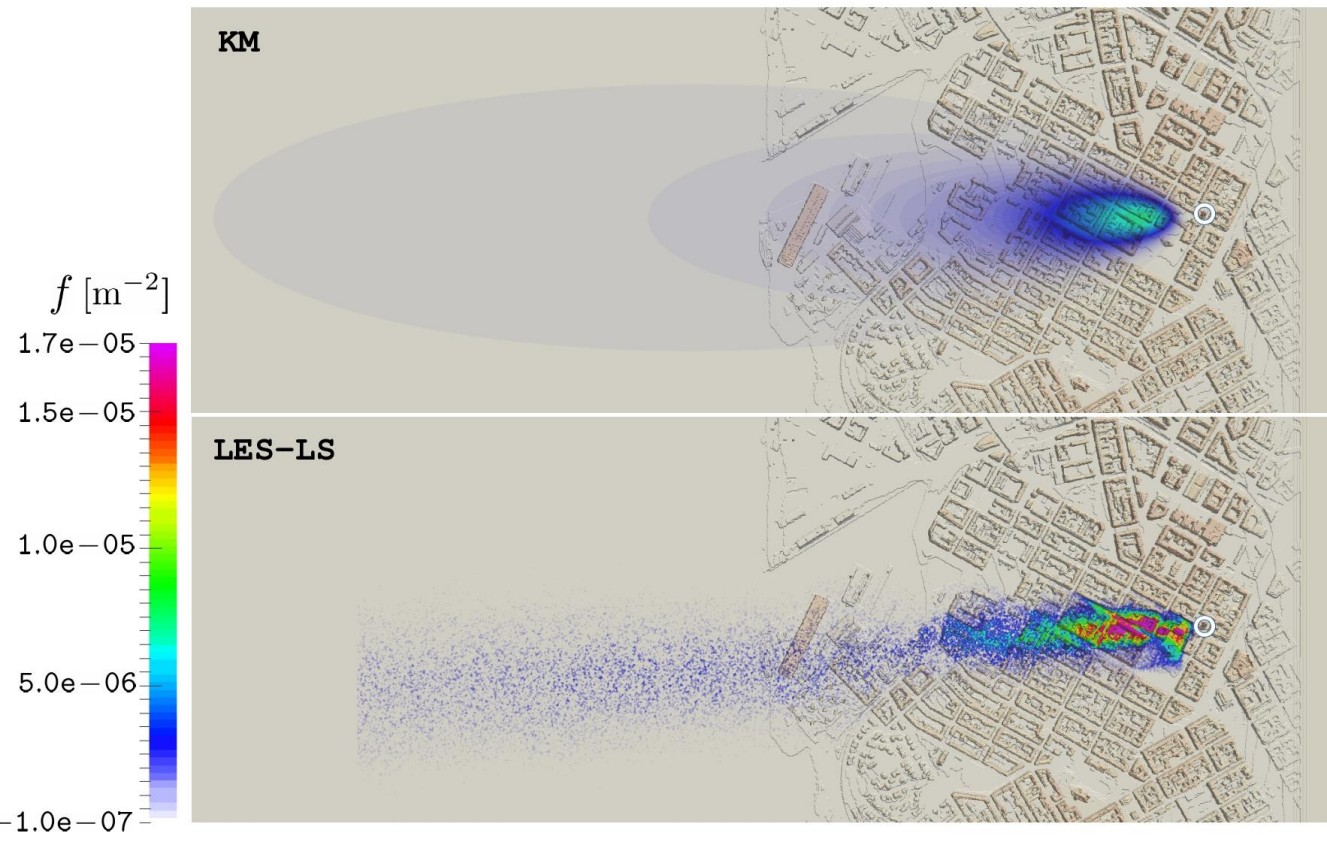

**Figure 8.** Comparison of identically normalized LES-LS (below) and KM (above) footprint distributions merged with the urban topography model of Helsinki. The location of the EC sensor (Hotel Torni) building is indicated with a white circle.

simplification the description of a measurement $\eta$ in Eq. (1) can be decomposed as:

$$\eta(\boldsymbol{x}_{\mathrm{M}}) = \sum_{e=0}^{N_{LC}} \eta_e(\boldsymbol{x}_{\mathrm{M}}) = \sum_{e=0}^{N_{LC}} \int_{\Omega_e} f(\boldsymbol{x}_{\mathrm{M}}, \boldsymbol{x}') \langle Q_e \rangle \ d\boldsymbol{x}' \tag{11}$$

where $N_{LC}$ is the number of different land cover types in the dataset and the constituents of $\eta$ are given by

$$\eta_e(\boldsymbol{x}_{\mathrm{M}}) = \int_{\Omega_e} f(\boldsymbol{x}_{\mathrm{M}}, \boldsymbol{x}') \langle Q_e \rangle \ d\boldsymbol{x}' = \langle Q_e \rangle \int_{\Omega_e} f(\boldsymbol{x}_{\mathrm{M}}, \boldsymbol{x}') \ d\boldsymbol{x}'$$

$$= \langle Q_e \rangle A_e. \tag{12}$$

Here, $A_e$ is the footprint weighted surface area of the $e^{\mathrm{th}}$ land cover type and

$$\Omega_e = \int_{\Omega} \frac{LC_e}{|LC_e|} \ d\boldsymbol{x}' \tag{13}$$





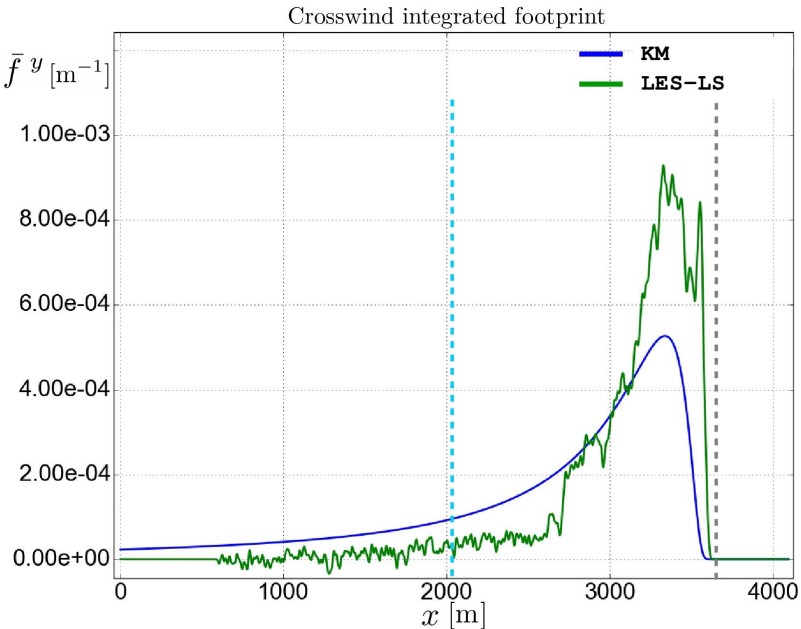

**Figure 9.** Comparison of normalized, crosswind integrated LES-LS and KM footprints. A light blue dashed line indicates the start of urban topography and the gray dashed line and the white circles mark the location of the EC sensor.

defines the corresponding subdomain that satisfy

$$\Omega = \sum_{e=0}^{N_{LC}} \Omega_e. \tag{14}$$

Now it is convenient to define two measures that facilitate a meaningful comparison between different footprints: The fractional contribution to the measurement from each constituent

$$r_e = \frac{\eta_e}{\sum_e \eta_e} \tag{15}$$

which require that $\langle Q_e \rangle$ are assigned for each land cover type, and the source-area fraction

$$a_e = \frac{A_e}{\sum_e A_e} \tag{16}$$

that provide an easy estimate of the footprint's coverage independent of source strength information (or assuming identical $\langle Q_e \rangle$ for all $e$). For proper assessment, these two fractions should be inspected in tandem.

The comparison is carried out by extracting the area corresponding to the LES domain from the $LC$ dataset, shown in Fig. 10, which has been modified to include the relevant streets in the vicinity of the footprint for the purpose of including the effect of traffic emissions into the demonstration. The obtained $f_{\mathrm{LES}}$ and $f_{\mathrm{KM}}$ footprints are then projected onto this raster map to compute the required integrals and fractions.





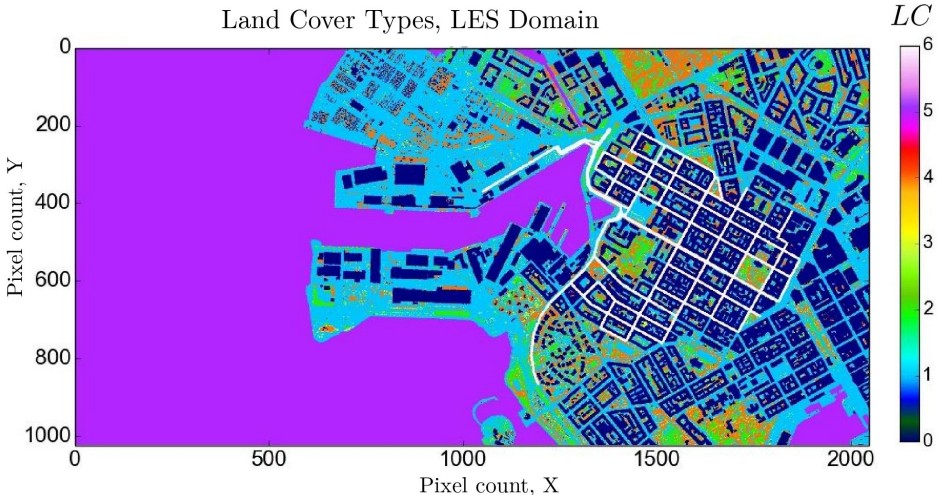

**Figure 10.** Raster map of land cover types, $LC$, within the LES domain. The original surface type classification data in Fig. 1 has been augmented by adding streets ($LC = 6$) to the relevant footprint area.

A pie-chart of source-area fractions $a_e$ from Eq. (16) for the Hotel Torni's flux footprint is demonstrated in Fig. 11, which provides an informative overview on the differences in source-area coverages. In this particular example, the analytical KM model gathers a significantly larger contribution from the far-field, which is reflected in the significantly higher coverage of water surface area. On the other hand, assigning each land cover type its corresponding – potentially fictional – mean source

strength $\langle Q_e \rangle$ and evaluating the fractional contributions $r_e$ from Eq. (15) provides means to carry out simplified virtual experiments concerning particular EC measurements. To demonstrate, consider $CO_2$ flux measurements in a situation where 95% of the $CO_2$ emissions originate from traffic (i.e. from roads) and 5% arise from other anthropogenic sources, which are emitted through ventilation outlets on the building roofs. For the sake of simplicity, the contibution from water area is assumed negligible and vegetation is assumed to act as a uniformly distributed sink over the land, which does not influence the ratio of

source contributions in the measurement. Utilizing an undefined reference source strength $\langle Q_{\mathrm{ref}} \rangle$, the sources are expressed as $\langle Q_e \rangle = \lambda_e \langle Q_{\mathrm{ref}} \rangle$, where the weights satisfy $\sum_e \lambda_e = 1$. Thus, in this example $\lambda_0 = 0.05$ for buildings and $\lambda_6 = 0.95$ for roads.

The measurement decomposition for this contrived situation is illustrated as a pie-chart in Fig. 12 which reflects how the interrelationship of the considered source-areas dictate the outcome. In this example, while the two footprints have distinctly different source-area fractions for buildings and roads, their ratios are close since $(A_6/A_0)_{\mathrm{LES}} = 1.1 \, (A_6/A_0)_{\mathrm{KM}}$ as seen in

Fig. 11, which is the reason for obtaining such comparable measurement decompositions.

Repeating the introduced assessment technique for multiple representative meteorological conditions, paves the way for a numerical approach that allows the obtained urban flux measurements to be interpreted either differently or with improved confidence. Naturally, having access to real source strength distributions opens up the ability to utilize LES-LS footprints (or positively assessed analytical footprints) to carry out detailed emission inventories, (e.g. Christen et al., 2011).




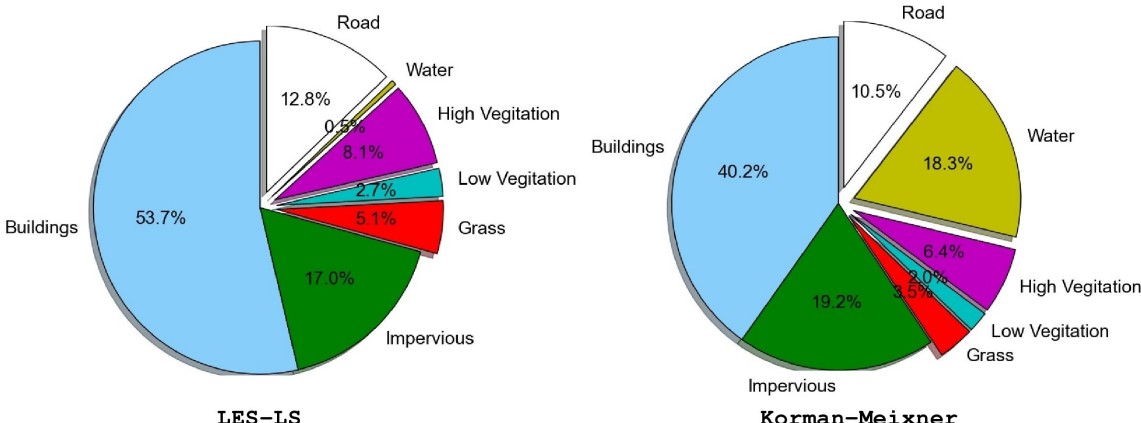

**Figure 11.** Comparison of source-area fractions $a_e$ resulting from applying $f_{LES}$ and $f_{KM}$ to the raster map of land cover types in Fig. 10.

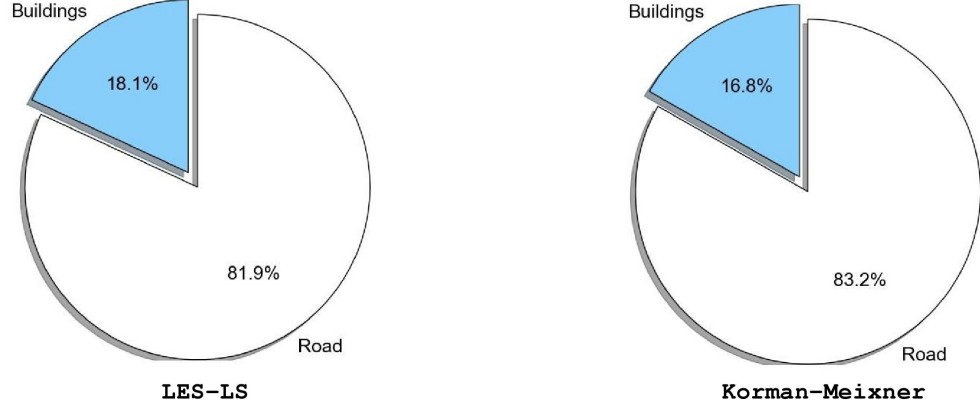

**Figure 12.** Comparison of a virtual $CO_2$ flux measurement decompositions $r_0$ and $r_6$. In this example, 5% of the $CO_2$ emissions are set to originate from anthropogenic sources (buildings) while the rest 95% stem from traffic (roads).

## 4   Summary and conclusions

The utility of eddy covariance method in measuring the exchanges of mass, heat and momentum between the urban landscape and the overlying atmosphere largely depends of the ability to determine the effective source-area, or footprint, of the measurement. In situations where the heterogeneity of the surface becomes relevant, like for urban landscape, and the structures surrounding the measurement site can no longer be considered as a homogeneous layer of roughness elements, the use of analytical footprint models becomes highly suspect. In order to diminish the resulting uncertainties and to obtain the ability to assess the applicability of analytical models, the ability to evaluate complex footprints with high resolution becomes essential.

This work presents a numerical methodology to generate topography-sensitive footprints for real urban EC flux measurement sites. This methodology is based on high-resolution LES-LS analysis where the simulation domain features a detailed



description of the urban topography and accounts for the entire vertical extent of the atmospheric boundary layer. The online-coupled LS model within the LES solver is employed to simulate a constant release of inert gas emissions from the potential upwind source-area of the considered EC sensor. The necessary data for the footprint generation is obtained from the LES-LS analysis by setting up a finite target volume around the sensor location and, over a sufficiently long simulation period, gath-

ering a record of particles that hit this target. The obtained dataset is subsequently post-processed to yield an estimate for the footprint, but if the considered EC sensor is mounted on a building instead of a conventional tower-like structure, standard post-processing techniques fail to produce a physically meaningful footprint. Therefore, this work introduces a robust piece-wise post-processing strategy, which facilitates the evaluation of the footprint despite the added complexity. The piecewise approach involves splitting the original dataset into a series of subsets which are all independently post-processed to yield

incompletely converged intermediate footprint estimates. Eventually, the final completely converged footprint is selectively assembled from the obtained set of intermediates.

The methodology is demonstrated in a real urban application where the objective is to compute a highly resolved topography-sensitive footprint for the SMEAR III EC flux measurement sensor mounted on the roof of a tall building situated in the downtown area of Helsinki, Finland. The EC sensor's measurement height is 60 m above the ground level and 36 m above

the surrounding mean building height (24 m). The meteorological conditions for the LES simulation were adopted from measurements on September 9[th] 2012 when south-westerly winds and a neutrally stratified boundary layer of 300 m height were recorded. A detailed topography map of Helsinki at 2 m resolution from Nordbo et al. (2015) was utilized to construct the topography model for the LES-LS domain. The resolution of the computational mesh was set at 1 m throughout the domain to ensure that the relevant turbulent structures even at level of street canyons were captured. An arbitrarily sized target box for

sampling the Lagrangian particle hits was setup around the sensor location, which collected ca. $19 \times 10^6$ particle hits during 3 hours of simulation time. The obtained dataset was subjected to the proposed piecewise post-processing method, demonstrating the functionality of the approach under various user-selected specifications. The obtained footprint stood in stark contrast to gradual ellipse shaped analytical footprints: The distribution exhibited strong adherence to the building block arrangement in the near field where the weight distribution changed abruptly between roof tops and street canyons. In comparison to the

Korman and Meixner (2001) (KM) model, the LES-LS footprint also exhibited stronger contribution from the near field, but more rapidly diminishing contribution from the far field.

This paper also introduces an accessible technique to employ the obtained high-resolution topography-sensitive urban footprint in estimating the potential error that may arise when an analytical footprint model is used to interpret urban EC measurements. The underlying stipulation for this method is that it does not require knowledge of real source strength distributions.

Thus, it is proposed that a detailed land cover type classification ($LC$) dataset is utilized as a model source strength distribution map for the urban surroundings assuming that it reflects the heterogeneity of the urban conditions sufficiently. Projecting a footprint distribution result onto such $LC$ map enables the evaluation fractional contributions, which indicate how each land cover type is represented in the measurement. This procedures provides a comparative technique to assess the effective deviations between different footprints. The demonstrated comparison between the LES-LS and analytical KM footprints in the

EC measurement setup in Helsinki revealed substantial differences in the fractional contributions when all land cover types





are considered equally relevant. However, when the effective source-area was limited to only roads and buildings to mimic a situation for $CO_2$ flux measurements, the fractional contributions became closely comparable despite the striking differences in the footprints.

The context of this paper is limited to laying out the new methodology for generating urban footprints and exploiting them in the assessment of analytical models. It is evident that changes in the meteorological and anthropogenic conditions will influence the results and a proper assessment of the applicability of analytical models at a given EC measurement site will require that these conditions are varied necessitating numerous footprint evaluations. This paper lays the numerical groundwork for such future investigations.

## 5   Code availability

PALM is an open source software released under GNU General Public License (v3) and freely available upon registration at https://palm.muk.uni-hannover.de/trac. The source code for handling the target box particle data acquisition is available by request from the corresponding author. The Python scripts used for the topography raster map manipulations and footprint post-processing and analysis are part of a larger library developed and maintained by the author. Currently access to the code repositories is granted by request only. Python is an open source programming language, which is freely available at www.python.org and www.numpy.org. The visualizations are performed with ParaView, an open-source, multi-platform data analysis and visualization application which is freely available at www.paraview.org.

*Acknowledgements.*  This study was supported by Academy of Finland (grant no. 284701, 1281255, 277664, and 281255). The computing resources were provided by CSC - IT Center for Science Ltd., Finland (grand challenge project gc2618). The author would like to sincerely acknowledge Curtis Wood for the meteorological data acquisition and Tiina Markkanen, Sigfried Raasch, Andrey Glazynov and Juha Lento for the help and advice they provided.



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
