# Peer review of "Numerical Framework for the Computation of Urban Flux Footprints Employing Large-eddy Simulation and Lagrangian Stochastic Modeling"

_Geoscientific Model Development, 2016_

## Author Comment (AC1) · 5 Jan 2017

There is a typo in the acknowledgements. Siegfried appears as Sigfried. This will be corrected by the author in the next revision.

---

## Short Comment (SC1) · 20 Jan 2017

Dear authors,

in my role as Executive editor of GMD, I would like to bring to your attention our Editorial version 1.1:

http://www.geosci-model-dev.net/8/3487/2015/gmd-8-3487-2015.html

This highlights some requirements of papers published in GMD, which is also available on the GMD website in the 'Manuscript Types' section:

http://www.geoscientific-model-development.net/submission/manuscript_types.html

In particular, please note that for your paper, the following requirements have not been met in the Discussions paper:

- "The main paper must give the model name and version number (or other unique identifier) in the title."

In order to simplify reference to your developments, please add a name (and/or its acronym) and a version number for the numerical framework in the title of your article in your revised submission to GMD.

Yours,

Astrid Kerkweg

---

## Referee Comment (RC1) · Anonymous Referee #1 · 22 Feb 2017

General considerations

In this contribution the authors present a methodology to calculate footprints in urban areas based on (very) high-resolution large eddy simulation (grid resolution 1 m) in combination with Lagrangian stochastic particle modeling. The need for this endeavor is motivated by stating that the usually employed analytical footprint models cannot be expected to yield useful results in urban areas. This is a more-than-valuable argument – despite the fact that there are less simplified options than analytical models and despite, more importantly, the astonishingly good performance (at least in the judgment of the present reviewer) of such an analytical footprint model, see major comment below. Due to the very high resolution employed in their modeling approach, the novelty

of their approach and some of the assumptions (see below) the authors encounter a number of problems ('numerical difficulties' (p6, l.7) to be worked around, or negative footprints (Fig. 5) to be avoided, etc.). These are all addressed and for each a 'solution' is presented - that can be defended (and is somehow defended by the authors), but is not in every case very obvious or unique. Still, the authors treat their results 'as the truth' (Section 3, see major comments 6, 7 and 8) rather than addressing the sensitivity of their results to their assumptions. I consider this to be a quite timely and important contribution to the problem of urban footprint modeling – but given the missing verification (which is indeed very difficult to achieve, as the authors state) I suggest to focusing more strongly on the sensitivity of the obtained results to the assumptions made, rather than presenting them as the final solution. In the major comments below I point to some of these assumptions.

Major comments

1) Quite influential treatment of the building - called 'topography' - model, (p6, points 1-3) and the flow simulation (precursor simulation with slip condition at the top, some sea surface roughness assumption, I presume, etc.). Probably, these are the result of some testing, e.g. point 3 of the building model. Can the authors comment on the impact on the simulated flow characteristics – and thus realism?

2) Release of particles 1 m 'above topography' (p8, l. 16): with the use of the word 'topography' in this paper, this means that surface emissions (traffic, say) and roof-top emissions (e.g., domestic heating) are treated to yield the same 'source' height'. In other words: the footprint function is clearly a function of all three spatial coordinates; if the height is defined in the way the authors do, this implies that '1 m above street' and 1 m above roof' (are treated to) experience the same physical processes, despite fact that one is indeed close to a solid surface below while the other is situated 'in the middle of the roughness sublayer', and possibly located above a slated roof. This refers to a very specific understanding of 'surface' that is distinctly different from other possible treatments (e.g., treating the roughness elements a porous surface) which is

possibly defendable – but has to be explicitly defended.

3) Size of sensor box: all the reasoning on p9 is understandable but at the same time the sensor box's size is quite arbitrarily chosen. Why is it 8 m in x, 20 m in y and 12 m in z (especially the latter choice is crucial!)? How sensitive are the results on these choices? How do these dimensions compare to the mentioned dominant scales of turbulence in the given example? Do the authors claim that these are general relations? Also, the authors state that "it pays off to obtain ... a large dataset accepting that it contains certain percentage of particle hits whose contribution will be discarded." (p9, l. 27) How (based on what) will certain particles be discarded? How can this be justified?

4) Mean vertical wind and associated 'far field correction' (p13, l. 35): quite some effort is dedicated to produce some 'plausible results' (no negative footprints). One would assume that the predominantly negative far field footprint results (as the authors state, 'due to the coordinate rotation', p13, l.25) from the local flow deformation (and associated gradients) in the vicinity of the target. In experimental work this issue is addressed with filtering of the data (trend removal, running mean) or applying what is called a 'planar fit' (Wilczak et al. 2001) – sometimes with different planes for different approach flow sectors etc. In the present framework this would (approximately) correspond to reducing the sizes of the sub-boxes of the target volume. Apparently, this did not work out (p13, l. 5) due to the restriction in the number of particles, but nevertheless it would be interesting to learn up to which discretization this has been tried (and with which results). Also, maybe a smaller number of particles (in a sub-box) would suffice to obtain an overall (spatial) trend of mean vertical velocity? Can the authors comment on any of these (apparently performed) trials and tests? The proposed far field correction – while apparently producing useful results in this very case – appears to be anything but general: for example if for another target box the resulting far field would be (slightly) positive, it would be quite difficult to argue that this would be removed by the correction.

5) There are a number of quite subjective judgments and choices. These include

- for example the number of required particles that is mentioned several times, but never substantiated.

- the size of the sensor box (major comment 3)

- p16, l. 25: 'encompassing only the near field', i.e. ca. 30 % of the total length of the LES domain. Why 30%?

- p16, l.30 : 'in this case study the threshold was set to include f(3)...' Why 3? What is the consequence of this choice?

6) The authors claim to have developed a 'technique' to estimate the error of a simpler, analytical footprint model when applied over an urban area. The technique, however completely relies on the assumption that their footprints are correct. Given the quite subjective assumptions they need (see previous comment), this assumption may not necessarily be very good. This 'technique' should therefore rather be labeled as a sensitivity test, thus avoiding to claiming the own results to be 'correct' when this cannot be demonstrated. It may also be noted that the similarity of the cross-wind integrated footprints (Fig. 9) is striking – given the simplification in the KM approach. On the other hand, the KM based much larger cross-wind dispersion (Fig. 8) is largely due to the larger (relative) source height. This should also be mentioned (see major comment 2).

7) Further on assessing differences between LES and KM footprints (Section 3.1, specifically p22, l.10ff): '....which has been modified to include the relevant streets...': this is another example of a subjective choice (see major comment 5): why are not all streets in the domain included? Can the authors comment on this? What impact would it have on the KM results?

8) The 'CO2 example' is extremely non-conclusive since it includes so many additional, and also not explained, assumptions (no impact of water sources, vegetation is uniformly distributed over land and has the same height [even if there are 'high vegetation' and low vegetation' areas], etc., etc.). The very same conclusion could have been

obtained by changing some parameters in the KM model alone (e.g., the roughness length being 10% of the mean building height). It is suggest to completely remove this entire section.

Minor comments

P2, l.2 such a sensor's. . ..

P2, l.10 'topography' is usually employed in connection with landscapes (hills, etc.) while here and in the following (apparently) a 'building topography' is referred to. To avoid misunderstanding either the wording should be changed (throughout) or the use of this expression should be made explicitly clear at this early stage.

P2, l. 17 of the turbulent flow field

P2, l.23 measurements cannot be 'extracted'

P3, l.19 'just above the roughness sublayer': at 2.3 m above the nearest building, this statement cannot be true, when taking the definition of the roughness sublayer as a reference (e.g., Raupach et al. 1991). These authors define "The term 'Roughness Sublayer' will indicate the entire layer dynamically influenced by length scales associated by roughness elements. . ..". Clearly, any flow property at 2.3 m above a roughly 60 m tall building will be locally influenced. Can the authors comment on this?

Fig. 2 caption: please mention that this is the urban grid. For better understanding of the turbulence recycling approach it would probably be helpful to indicate also the precursor grid.

P6, l.10 EC measurements IN Torni? The authors probably mean on top of the Torni building.

P6, l.13 by means of . . .

P7, l.4 with a constant value

[Figure]

P8, l.10 the release of particles is activated: one would need to know how many particles are released (per time step, say and (probably) per grid cell.

P9, l.5 I don't think 'fixating on the exact location. . .' is very clear – please reformulate

P10, l.2 reference simulation is run for 1 h 'to develop ABL turbulence sufficiently' (what is sufficiently?) and averaging is performed for the last 45 minutes. In other words, the effective spin-up time is 15 minutes. How does this compare to some eddy turnover time for the given situation? Can the authors comment?

P10, l. 14 to what do the computational costs amount in absolute terms?

P11, l.2 'each lth particle's coordinate. . ..': as this is formulated, not every particle is sampled (conditional on its position within the box) but only every lth particle. Is this what the authors want to say? And if so, what is 'l' and how is it determined? What is the reasoning not to sample all particles but a subset equally spaced in l?

P12, l.3 how large is delta_xf chosen?

P13, l.1 close to the top

P13, l. 12 what is 'negative far field'? Negative vertical velocity in the far field? Or negative footprint in the far (upwind) field)? Please specify.

P15, l.10 the criterion

P16, l.25 which is not sufficient. . .: based on what?

Fig 7, caption: incomplete (caption must include all information to understand the figure without reading the text).

P19, l. 30 . . .which would otherwise be employed: how can the authors state this? There are many different models of this kind, so this is only one of the models that possibly might be used.

Fig.9 caption: I don't think I can see any white circles. . ..

P22, l.1 that leads to

Fig. 11 the entries (labels) 'vegitation' (high and low) probably mean to refer to 'vegetation' and should be changed.

References

Raupach MR, Antonia RA and Rajagopalan S: 1991, 'Rough-Wall Turbulent Boundary Layers', Appl. Mech. Rev., 44, pp 1-25

Wilczak JM, Oncley SP, Stage SA: 2001, Sonic anemometer tilt correction algorithms. Boundary-Layer Meteorol 99:127–150. doi:10.1023/A:1018966204465

———————————————

---

## Referee Comment (RC2) · Anonymous Referee #2 · 5 Mar 2017

Review of "Numerical framework for the computation of urban flux footprints employing large-eddy simulation and Lagrangian stochastic modelling" by M. Auvinen, L. Jarvi, A. Hellsten, U. Rannik, and T. Vesala

This is a worthwhile study and paper, and I support the approach of using a high fidelity LES and Lagrangian stochastic (LS) model to determine footprints in an urban area. I have two main comments and several detailed ones for the authors to address, but overall believe that the paper should be published subject to the attention of the comments.

**Main Comments**

1) The first issue centers on: How well can one determine the $\overline{w}$ at the target volume and the uncertainty associated with it since this affects the uncertainty in the footprint through Eqs. (4) and (5). The authors argue that a well-defined value of $\overline{w}$ cannot be determined at the target site for reasons discussed on pages 9 (lines 10 - 21) and 13 (lines 24 - 33), and that this limitation leads to the negative bias in the far-field footprint (Fig. 5). They go on to develop the far-field correction approach (pages 13 to 15), which involves a correction coefficient $c_{ijk}$ applied to the original $\overline{w}$.

I believe that the authors should provide further information to quantify the uncertainty in $\overline{w}$ at the target site (volume) and also upstream of it. In particular, they should compute and provide: $\overline{w}$, $\sigma_w$, and the uncertainty in $\overline{w}$ (in a statistical sense) from their numerical wind field data: 1) over the urban landscape upstream of the target site but at the same height (or over the same heights) as the target volume [an areal average over the upstream domain or volume using a few heights to match those of the target volume and extending from the leading edge of the city, $x \simeq 2$ km (Fig. 2), to the Hotel Torni], and 2) over the target volume. The difference in the raw $w$ values over the target volume (larger) and upstream area (smaller) presumably should lead to a larger $\overline{w}$ and greater variance ($\sigma_w^2$) over the target volume, fewer number of numerical data points, and greater uncertainty in $\overline{w}$ over the target volume (e.g., uncertainty estimated as some factor $\lambda$ depending on the $w$ PDF times $\sigma_w/\sqrt{N}$, $N$ being the number of points).

Furthermore, it would be useful to compute the subgrid-scale (sgs) turbulent kinetic energy (TKE, $E_s$) and the LES resolved scale TKE ($E_r$), and their ratio or an sgs rms velocity, $\sigma_{sgs} = (2/3)E_s$ and the ratio, $\sigma_{sgs}/\sigma_{wr}$. This ratio would quantify the importance or not of the sgs velocities in determining the footprint; as with the $\overline{w}$, $\sigma_w$, etc., this should be done for both the upstream area/volume and the target volume to make the distinction if there is one.

2) Section 2.3.2. This section is one of the more limiting parts of the paper in being somewhat arbitrary and subjective for generating the final LES-LS footprint. I believe that it is sufficient as an initial approach/procedure. However, for use by other researchers or as a standalone method for generating footprints (by others) for multiple EC sites in a city, a more rigorous, robust, and less subjective approach is necessary. For example, on page 16, is the $f^{(0)}$ with $4 \times 10^5$ particle entries deemed insufficient as a

reference footprint, $f^{REF}$, because there is too much blue (negative footprint value) in the far field of the footprint (Fig. 6)? Conversely, is $f^{(1)}$ judged to be sufficient and act as the reference because there is less blue? There should be some quantitative method, index, or variable that stipulates the adequacy of the reference (and why); e.g., requiring the far-field negative fooprint to be less than some small fraction of the maximum footprint. Such a criteria is used in deciding whether a candidate footprint, $f^{(k)}$, should be included in the final assembly (page 16, lines 31 - 33) with the $||\Delta f||_{max}$ "determined according to the case-specific requirments." These requirements should be spelled out. However, the maximum values, $f^{(k)}_{max}$, associated with these difference maxima, $\delta f_{max}$, and the fraction, $\delta f_{max}/f^{(k)}_{max}$, should be given. The $||\Delta f||_{max}$ here was chosen from the $f^{(3)}$ value, but it is not clear why. Also, the maximum footprint values in Fig. 6 appear to be about $10^{-6}$ (bright red), but the $||\Delta f||_{max} = 6.24 \times 10^{-3}$ is much larger. I don't understand this.

**Detailed Comments**

1) page 2, line 4. "...which relates the value of a measurement (of flux or concentration) at location ..." The parenthetical words would make this clearer.

2) page 2, Eq. (1). It should be made clear here that the footprint, $f(\mathbf{x}_M, \mathbf{x}')$, has the dimensions inverse of the integration unit(s). In Eq. (1), this is just length ($L$) since the integration is along a line in the domain, e.g., $f$ could be the crosswind-integrated footprint, $f^y$, used later. But often, the footprint pertains to an elemental area $dxdy$ ($L^2$) as in Eq. (4).

3) page 3, lines 8, 9. "...conduct tracer gas experiments, which are nearly impossible to arrange in residential areas." However, this computational framework could be tested in some way in a large wind tunnel such as WOTAM at the University of Hamburg (e.g., Leitl and Schatzmann, 2010). This tunnel with dimensions of 4m × 3.2m × 25m (width, height, length) has been used to study many aspects of dispersion in large European cities. I would recommend that testing of the LES-LS numerical Helsinki footprint model be tested in some way in such a tunnel. It would give greater credibility to the approach especially when using the approach to suggest/demonstrate limitations in analytical footprint models.

4) page 11, line 11. The authors talk generally about the target subvolumes, size, and number, but this should all be guided the size/scale of the turbulent flow structures at the EC sensor site. This was mentioned earlier in the paper, but no specifics were given. For example, on page 9 (line 32), it is merely stated that "the target box reasonably represents the sensor site." Should the target box scale or total volume be of the order of or less than (say some fraction of) the characteristic flow structure dimension/volume at the EC site? If we assume very crudely that the characteristic length scale is $\ell_c = \kappa z$, with $\kappa$ being the von Karman constant (0.4) and $z$ is height, $\ell_c = 24$ m; i.e., for a neutral boundary layer over homogeneous terrain. The characteristic dimension of the sampling volume is $\ell_{sc} = (\ell_{sx} \cdot \ell_{sy} \cdot \ell_{sz})^{1/3} = 12.4$ m, which is half of $\ell_c$. I don't know the logic of choosing the $\ell_{sc}$ in the paper, but requiring it's overall dimension to be less or much less thant $\ell_c$ may be a useful criterion. Clearly, there may be others.

5) page 12, line 19. "distributions" Do you mean individual "footprint" distributions?

6) page 14, line 3. "negligbly small offset." But that offset is 20% to 25% of the maximum $\overline{f}^y$ in Fig. 5, and thus is not small.

7) page 15, Eq. (9). It might be noted that a similar approach is often used in determining the vertical tilt axis of a sonic anemometer on a meteorological tower; i.e., due to the mounting imprecision and/or possible variability in the sensor vertical axis over time (wind loads, vibrations, etc). Assuming that over a long time record above flat terrain the $\langle \overline{w} \rangle = 0$, the tilt axis ($\mathbf{c}$ or $c_{ijk}$) (relative to the 3 fixed coordinates) is chosen by ensuring $\sum_n^N \mathbf{c} \cdot \mathbf{u}_n = 0$ (sum of the vertical velocity components is zero), where $n$ is the measurement (realization or time) index and $N$ is the total number of measurements.

8) page 20, lines 8 - 29, Fig. 9. Why is the numerical footprint, $\overline{f}^y$, so much more peaked than the analytical Korman and Meixner (KM) (2001) model? There are at least two potential reasons for this as suggested by the authors. 1) The real urban terrain/fetch upstream of Hotel Torni is only about 1600 m or about 39% of the total LES domain (4096 m; Fig. 2). The real terrain which is heterogeneous with a marked change in roughness at the upwind edge of the city is accounted for in the LES, whereas the KM presumably assumes homogeneous conditions upstream. These very different mean wind and turbulence fields could possibly be accounted for in the KM model in a crude way (point 9 below). 2) The mean wind and rms lateral turbulence component ($\sigma_v$) for the KM model were extracted from the measurements at the top of the Hotel Torni. The authors should repeat the KM calculation using the LES mean wind and $\sigma_v$ at the height of Hotel Torni. This could be done using both the LES values 1) within the target volume and 2) the upwind $U$ and $\sigma_v$ values. That is, there should be consistency in the meteorological inputs to the LES-LS and the KM model.

9) For future development of the KM or other analytical footprint models, it would be interesting to run KM or other model for an $x$-direction terrain inhomogeneity, sea-to-land, with a large increase in roughness at the interface to see if that would give a zeroth order change/improvement in the analytical footprint. [Note, this is not the authors' responsibility and need not be addressed by them.] Horst and Weil (1994) and perhaps others have shown how a step change in the surface flux can be accomodated. Here, however, it would be necessary to account for a change in the mean wind and turbulence profiles at the land-sea interface. This may be possible to do in a simple way (parameterization) based on an internal boundary layer model characterized by a sudden roughness change (and/or heat flux) (e.g., Brutsaert, 1984; Garrett, 1992) where new wind and turbulence profiles can be developed downwind of the change.

**Additional References**

Brutsaert, W., 1984: *Evaporation into the Atmosphere*, Reidel Publishing Co., Dordrecht, 299 pp.

Garrett, J.R, 1992: *The Atmospheric Boundary Layer*, Cambridge University Press, 316 pp.

Horst, T.W., and J.C. Weil, 1994: How far is far enough?: The fetch requirements for micormeteorological measurement of surface fluxes. *J. Atmos. Oceanic Tech.*, **11**, 1018–1025.

Leitl, B., and M. Schatzmann, 2010: Validation data for urban flow and dispersion models — are wind tunnel data qualified? Presented at: Fifth Symposium on Computational Wind Engineering (CWE2010), Chapel Hill, North Carolina, USA, May 23–27, 2010.

---

## Author Comment (AC2) · 8 Jun 2017

In the guidelines for Development and Technical Papers, it states the following: *If the model development relates to a single model then the model name and the version number must be included in the title of the paper.*

As the technical developments in the manuscript examine modelling processes that require joint operation of **two** models (PALM and P4UL), we wish to propose that the term 'Numerical Framework' would be more appropriate in the title. The model names, version numbers and accessibility information will be detailed in the Code Availability section.

---

## Author Comment (AC3) · 9 Jun 2017

The developments described in the manuscript do fall somewhat between the 'Development and Technical' and 'Model Description' paper type categories. But, upon reviewing the paper type descriptions carefully, we are inclined towards 'Development and Technical Papers' because the statements referring to *"technical developments relating to model improvements"* and *"technical aspects of running models"* in the paper type description best characterize the content of our manuscript (although this is clearly dependent on our interpretation of the word 'technical' in this context). Thus, depending upon the editors approval, we would prefer to leave the paper type unchanged.

---

## Author Comment (AC4) · 9 Jun 2017

We have improved the Code Availability section by adding a DOI for the pre- and post-processing library P4UL, which we have now made freely available under the MIT license. Also, in accordance with the guidelines of GMD, the version and revision number of PALM LES model has been added as well.

---

## Author Comment (AC5) · 26 Jun 2017

The comment was uploaded in the form of a supplement:
https://www.geosci-model-dev-discuss.net/gmd-2016-302/gmd-2016-302-AC5-supplement.pdf

---

## Author Response (AR1)

**Response to Referee 1.**

**Mikko Auvinen, Leena Järvi, Üllar Rannik, Antti Hellsten, Timo Vesala**

June 26, 2017

**Referee #1 comments**

All page and line numbers refer to the *original* manuscript.

- (1) Referee comment: Quite influential treatment of the building called 'topography' model, (p6, points 1-3) and the flow simulation (precursor simulation with slip condition at the top, some sea surface roughness assumption, I presume, etc.). Probably, these are the result of some testing, e.g. point 3 of the building model. Can the authors comment on the impact on the simulated flow characteristics and thus realism?
- Author's response: It is our understanding that the precursor driven initialization and turbulence recycling inlet boundary condition setup represents the state-of-the-art when considering non-periodic atmospheric boundary layer flows. We've considered in our tests many different ways to compute the periodic precursor run: for instance, driving the flow with a mean pressure gradient to a quasisteady solution or initializing the system with a desired amount of momentum and running the computation with a mass conservation option on until a desired turbulent boundary layer profile evolves. In this study, we wanted to fix the boundary layer height to a measured value ( $\delta \approx 300$  m) by specifying a potential temperature profile with a strong inversion. This necessitates the use of the latter approach. The  $\sim 150$  m buffer zone between the inversion layer and the upper boundary allows also the larger ABL turbulent structures to develop and diminishes the effect of the upper boundary condition on the solution for  $z < \delta$ . This brings a level of realism to our urban ABL simulations that we have not observed elsewhere. The author has mistakenly reported the precursor's top boundary condition as a slip wall. (This stems from an unfortunate mistake of documenting, partially, the precursor settings belonging to a different application. This also relates to one of the minor comments below.) The top boundary condition in the precursor simulation is of type Dirichlet (fixed value at  $u_{top} = u_g$ ), while the top boundary condition in the main simulation is a slip wall.

In specifying the surface roughness for the urban simulation we followed the approach of Letzel et al. (2012), who utilized a uniform  $z_0 = 0.05$  m to qualitatively account for the aerodynamic effect of "moderately rough walls", which entail surface extrusions such as balconies, chimneys, ventilation ducts, stationary cars, small-scale vegetation etc. (i.e. structures not in the digital elevation model.) Admittedly, the chosen value is too high for the sea surface, but in its current state, PALM does not have the capacity to handle heterogeneous distributions of surface roughnesses. This functionality is currently under development.

The ramp just upwind of the outlet boundary is a result of a series of numerical experiments and has proven the most effective and computationally most efficient 'buffer zone' outlet treatment. In our experiments, no detectable effect was observed at the EC measurement site (Hotel Torni), which is situated  $\sim 300$  m upstream from the start of the ramp. According to our current knowledge, this could be further reduced to save computational cost.

Author's changes in manuscript: Corrected and augmented statements concerning the top boundary condition in the precursor and urban simulation were included on P7 l.1 and P8 l.4.

Statement declaring the roughness length and the associated citation to Letzel et al. (2012) is added on P7 1.14.

- (2) Referee comment: Release of particles 1 m 'above topography' (P8, l. 16): with the use of the word 'topography' in this paper, this means that surface emissions (traffic, say) and roof-top emissions (e.g., domestic heating) are treated to yield the same 'source' height'. In other words: the footprint function is clearly a function of all three spatial coordinates; if the height is defined in the way the authors do, this implies that '1 m above street' and '1 m above roof' (are treated to) experience the same physical processes, despite fact that one is indeed close to a solid surface below while the other is situated 'in the middle of the roughness sublayer', and possibly located above a slated roof. This refers to a very specific understanding of 'surface' that is distinctly different from other possible treatments (e.g., treating the roughness elements a porous surface) which is possibly defendable but has to be explicitly defended.
- Author's response: This is indeed a difficult topic considering all the possible emission scenarios. In this context, setting the particle emission height to 1 m above all surfaces is fundamentally a modeling decision which is, first, motivated by the desire to take into account both traffic and anthropogenic sources while avoiding the clustering of too many particles within the first cell layer and, second, influenced by the topography model's level of detail and the used LES resolution. We've observed during this and a preceding study by Hellsten et al. (2015) that setting the source height at 1 m above solid surfaces allows the footprint to adjust to the topographic details of the domain (at 2 m resolution) sufficiently.
- Author's changes in manuscript: Added a specification (P8, 1.20) that the release height is set 1 m above solid surfaces.
- (3) Referee comment: Size of sensor box: all the reasoning on P9 is understandable but at the same time the sensor box's size is quite arbitrarily chosen. Why is it 8 m in x, 20 m in y and 12 m in z (especially the latter choice is crucial!)? How sensitive are the results on these choices? How do these dimensions compare to the mentioned dominant scales of turbulence in the given example? Do the authors claim that these are general relations? Also, the authors state that "it pays off to obtain ... a large dataset accepting that it contains certain percentage of particle hits whose contribution will be discarded." (p9, 1. 27) How (based on what) will certain particles be discarded? How can this be justified?
- Author's response: The methodology is fundamentally constructed such that there are no strict conditions for determining the initial size of the sensor box as the final 'effective' sensor box is ultimately determined through the selective assembly procedure. Thus, the final result is a function of a set of subvolume contributions and does not directly depend on the initial size ... given that it contains a sufficiently large dataset of particle hits. (To clarify that the discarding of certain particle hits – and the associated justifications – take place in the selective assembly phase, a statement referring to the appropriate section is added.) It is difficult to dictate apriori what dimensions  $\Delta x_{\rm T}$ ,  $\Delta y_{\rm T}$ ,  $\Delta z_{\rm T}$ should optimally obtain and, for this reason, the authors are very careful not to make general or exclusive statements about the initial size of the sensor box. But, we do agree that some guidance should be provided and, in the Hotel Torni example, the horizontal length scale  $L_{\rm T}$  of the building's apex structure does function as a useful reference scale.

We do not wish to make any general statements about the relation between the initial sensor box size and the relevant turbulent scales. But, referring to the author's response to Referee comment (4) below, it has now been uncovered that, in the presence of strong mean flow gradients, the conventional coordinate rotation requires that the *subvolume* size adheres to the LES grid cell size. Thus, it can be stated that the strength of the mean flow gradients dictates the dimensions of the subvolumes and, under the presented circumstances (i.e. in the vicinity of a building roof), these dimensions become identical with the resolved scale of the LES simulation.

- Author's changes in manuscript: On p9 l.24-29 the sensor box dimensions are expressed in terms of the horizontal length scale  $L_{\rm T}$  and on p9 l.28 the statement referring to the section where selective assembly is employed to discard certain particles is added for clarification. See also the changes related to the comment (4) below.
- (4) Referee comment: Mean vertical wind and associated 'far field correction' (p13, l. 35): quite some effort is dedicated to produce some 'plausible results' (no negative footprints). One would assume that the predominantly negative far field footprint results (as the authors state, 'due to

the coordinate rotation', p13, l.25) from the local flow deformation (and associated gradients) in the vicinity of the target. In experimental work this issue is addressed with filtering of the data (trend removal, running mean) or applying what is called a 'planar fit' (Wilczak et al. 2001) sometimes with different planes for different approach flow sectors etc. In the present framework this would (approximately) correspond to reducing the sizes of the sub-boxes of the target volume. Apparently, this did not work out (p13, l. 5) due to the restriction in the number of particles, but nevertheless it would be interesting to learn up to which discretization this has been tried (and with which results). Also, maybe a smaller number of particles (in a sub-box) would suffice to obtain an overall (spatial) trend of mean vertical velocity? Can the authors comment on any of these (apparently performed) trials and tests? The proposed far field correction - while apparently producing useful results in this very case - appears to be anything but general: for example if for another target box the resulting far field would be (slightly) positive, it would be quite difficult to argue that this would be removed by the correction.

Author's response: Initially we experimented with trend removal in the hopes it would remove the strong bias from the footprint, but the effects were insignificant. We experimented with different target volume discretizations, but here we were mentally oriented to think that the vertical discretization is of primary interest because, at the limit, the subvolumes should approach plains across which particle intersections are monitored in accordance to the footprint evaluation technique used, for instance, in (Hellsten et al., 2015). As it turns out, this line of reasoning was misleading. All the earlier experiments where the emphasis was placed on varying  $n_z$  resulted in consistently negative far fields. Refining the target box discretization (by primarily increasing  $n_z$ ) yielded consistently negative asymptotes, but with decreasing magnitude. However, the reduction in magnitude occured at a rate that lead us to conclude that the required level of discretization is not feasible. (It's also worth noting that, from the beginning there was an underlying objective to obtain a partically converged, but nonetheless meaningful distributions for each individual  $f_{i,i,k}$  to facilitate the selective assembly.) However, now prompted by the comment, the author decided to pursue this issue once more and approach the problem without the presupposition that the targets should approach a plane or that each sectional result should be an 'inspectable' distribution. The new experiments uncovered what now seems rather obvious: When the target box is discretized in accordance with the LES grid (with uniform 1 m resolution), the coordinate rotation becomes successful and the far field of the final footprint approaches the anticipated zero asymptote. This process entailed generating  $n_x \times n_y \times n_z = 8 \times 20 \times 12 = 1920$  subvolume contributions which is an excessive operation, whereas coarsening the discretization to 2 m (and genarating 240 subvolume contributions) immediately introduced the negative bias again.

This is indeed significant and – putting aside the unreasonable post-processing of thousands of  $f_{i,j,k}$  contributions – establishes that the piecewise processing can be utilized without any far field correction in situations where the conditions around the sensor site are problematic such that the mean flow field exhibits strong gradients and assumptions concerning well-mixed conditions for the particles cannot be warranted. However, under such circumstances, this approach comes with the requirement that the discretization of the target box closely adheres to the discretization of the LES model, which leads to an overwhealming number of degrees of freedom in the post-processing phase and particularly in the selective assembly step. Thus, the strong motivation for the far field correction approach remains. But, this result provides an excellent reference that was previously missing for the far field correction strategy. Previously we relied on the outcome that the far field correction approach, completed at different target volume discretization. Now, the 'target-box-resolved footprint' assembled from subvolume contributions within  $\mathcal{V}_{\rm EFF}$ , which corresponds to the effective target box size determined by the selective assembly procedure performed with a coarser discretization, provides a convenient reference for the comparisons.

- Author's changes in manuscript: Section 2.3.1 on page 12-14 is now rewritten to reflect the clarity that the finely resolved piecewise processing provides. Figure 5. is redrawn to illustrate the effect of incrementally refined discretization of the target volume. The final footprint comparison in Fig. 7 is also redrawn to include the uncorrected reference result. Statements and modifications concerning the newly drawn Fig. 7 are added to the last paragraph of Section 2.3.2. on P18. Related modifications are included in Section 4.
- (5) Referee comment: There are a number of quite subjective judgments and choices. These include

- for example the number of required particles that is mentioned several times, but never substantiated. (1) the size of the sensor box (major comment 3)

- Author's changes in manuscript: Clarification according to Hellsten et al. (2015) is added to P12, 1.17.
- (5) Referee comment: (2) p16, l. 25: 'encompassing only the near field', i.e. ca. 30% of the total length of the LES domain. Why 30%?
- Author's response: First, there is a bad choice of wording: the length is referred to the total length of the particle release area (in the LES domain.)

The value (30%) is obtained by inspecting footprints integrals over the near field and determining its extent such that the integral encompasses approximately 50 % of the total footprint contributions:  $\int_{\Omega_{\star}} f d\vec{x} \approx \frac{1}{2} \int_{\Omega} f d\vec{x}$ . This allows the relevant deviations in the near field, which are caused by the small displacements between individual  $f_{i,j,k}$  contributions within the target box, to be relected in the measure for discrepancy.

- Author's changes in manuscript: The wording is changed and the reasoning for choosing the value is added to the text (P16, 1.16-21).
- (5) Referee comment: (3) P16, 1.30 : 'in this case study the threshold was set to include f(3)...' Why 3? What is the consequence of this choice?
- Author's response: The selective assembly a process that has two competing objectives: The final footprint should be composed from a sufficiently large number of particle entries (>  $10^6$  according to (Hellsten et al., 2015) to ensure the distribution is well converged even in the far field) while introducing a minimal level of discrepancy. What is the optimal compromize between the two is ultimately an objective choice which depends on the allowed level of tolerances in study and the available computational resources. The method allows the consequence of this choice to be inspected and a statement to this effect is added to the manuscript. Please, also refer to the author's response to Referee #2 comment (2).
- Author's changes in manuscript: The changes associated with the Referee #2 comment (2) address the issue raised here.
- (6) Referee comment: The authors claim to have developed a 'technique' to estimate the error of a simpler, analytical footprint model when applied over an urban area. The technique, however completely relies on the assumption that their footprints are correct. Given the quite subjective assumptions they need (see previous comment), this assumption may not necessarily be very good. This 'technique' should therefore rather be labeled as a sensitivity test, thus avoiding to claiming the own results to be 'correct' when this cannot be demonstrated. It may also be noted that the similarity of the cross-wind integrated footprints (Fig. 9) is striking given the simplification in the KM approach. On the other hand, the KM based much larger cross-wind dispersion (Fig. 8) is largely due to the larger (relative) source height. This should also be mentioned (see major comment 2).
- Author's response: In rewriting the Section 2.3.1 it is now carefully stated that the method is an approximation and associated with error whose magnitude and sensitivity must be established. It has been the authors' intention from the very beginning to communicate that approximate nature of the proposed approach. Hopefully, this is now made clearer than before. The original intention was to show that, despite the subjective choices made in the footprint post-processing, the discrepancies between the differently obtained LES-LS footprints came out very small and basically insignificant particularly when examined in juxtaposition to the discrepancies between LES-LS and analytical footprints. With the discovery of the 'excessively laborsome' reference footprint, the confidence in the obtained LES-LS results has increased as this result can, in a very limited sense, be considered accurate. But, we do agree that there are still strong assumptions embedded in the methodology and therefore any reference to claiming that the numerical results are 'correct' should be avoided.

Concerning the larger crosswind dispersion of the KM model and specifying the measurement height for the Hotel Torni's sensor. This is indeed a difficult guestion. Accounting the effect of the buildings (within the footprint) on the measurement height value is very risky because it has a strong impact on the total extent of the footprint itself. Furthermore, the changing ground level within the footprint also needs to be considered. In the presence of such complexity, we decided to define  $z_m$  by subtracting the displacement height from Hotel Torni's above-sea-level height, because vast majority of the potential source area remained close to the sea level. However, we've decided to reverse this decision and, instead, subtract the displacement height from the above-grould-level height value. This places the emphasis on the near field, which in the light of the LES-LS footprint results, is certainly justifiable within the urban canopy. This also improves (to a small degree) the juxtaposition between the KM and LES-LS footprints comparison.

- Author's changes in manuscript: Many of the associated changes are included in the rewritten Section 2.3.1. The parametrization of the KM model for the comparison is changed (P20, Table 3) and the associated graphs featuring KM footprint are redrawn.
- (7) Referee comment: Further on assessing differences between LES and KM footprints (Section 3.1, specifically P22, 1.10ff): '... which has been modified to include the relevant streets ...': this is another example of a subjective choice (see major comment 5): why are not all streets in the domain included? Can the authors comment on this? What impact would it have on the KM results?
- Author's response: All the streets with traffic that reside within the area that the footprints cover are included. The ship yacht at the shoreline (on a peninsula) only includes streets that are not accessible to normal traffic.

**Author's changes in manuscript: No changes proposed.**

- (8) Referee comment: The 'CO2 example' is extremely non-conclusive since it includes so many additional, and also not explained, assumptions (no impact of water sources, vegetation is uniformly distributed over land and has the same height [even if there are 'high vegetation' and 'low vegetation' areas], etc., etc.). The very same conclusion could have been obtained by changing some parameters in the KM model alone (e.g., the roughness length being 10% of the mean building height). It is suggest to completely remove this entire section.
- Author's response: The 'CO2 example' is only included for the purpose of demonstrating the method of computing fractional contributions  $r_e$  using different footprints. This is a very useful approach to examine how sensitive the spatial interpretation of a particular EC measurement is to the changes in the footprint. We are fully aware that, ultimately, it requires proper information about the source area, which we do not have at this point. But, we do think it's important to demonstrate the method (even though it is very simple). For this reason, we have attempted to use language that conveys the nature of this demonstration and propose to make it even clearer by adding terms like "consider CO2 flux measurements in a *hypothetical* situation" on P23, l.6. We agree that the result is extremely non-conclusive and, therefore, refrain from any kind of analysis. Also, the Fig. 12 draws too much attention to the result, so we willingly remove it entirely and report the fractional contribution values within the text only.
- Author's changes in manuscript: Relevant parts in Section 3.1 are rewritten and Fig. 12 is removed. Also in the summary (Section 4), the part concerning the example is rewritten.

**Minor comments**

(9) Referee comment: P2, l.2 such a sensor's ...

**Author's changes in manuscript: Corrected.**

(10) Referee comment: P2, l.10 'topography' is usually employed in connection with landscapes (hills, etc.) while here and in the following (apparently) a 'building topography' is referred to. To avoid misunderstanding either the wording should be changed (throughout) or the use of this expression should be made explicitly clear at this early stage.

Author's changes in manuscript: A clarifying sentence added to P2, l.11.

(11) Referee comment: P2, l. 17 of the turbulent flow field

Author's changes in manuscript: Corrected.

(12) Referee comment: P2, 1.23 measurements cannot be 'extracted'

Author's changes in manuscript: 'Extracted' changed to 'obtained'.

- (13) Referee comment: P3, 1.19 'just above the roughness sublayer': at 2.3 m above the nearest building, this statement cannot be true, when taking the definition of the roughness sublayer as a reference (e.g., Raupach et al. 1991). These authors define "The term 'Roughness Sublayer' will indicate the entire layer dynamically influenced by length scales associated by roughness elements. . ...". Clearly, any flow property at 2.3 m above a roughly 60 m tall building will be locally influenced. Can the authors comment on this?
- Author's response: In the context of reporting results from Hotel Torni's EC site, it is often claimed that the measurements are obtained in the inertial sublayer (e.g., Nordbo et al., 2013) and this is based on the commonly used definition for the roughness sublayer where the layer is estimated to be 2 times the mean building height of the surrounding area. But, we fully understand that the issue of determining the height of the roughness sublayer – particularly in the context of using a tall building as an EC flux tower – is complicated and somewhat vague as different non-exact classifications exists. For these reasons, and in the absence of conclusive evidence, we willingly remove the statement on P3, l.19. In fact, the footprint results obtained herein do suggest that the EC sensor above Hotel Torni is within the roughness sublayer if the Raupach et al. (1991) definition is used.

Author's changes in manuscript: Statement removed.

- (14) Referee comment: Fig. 2 caption: please mention that this is the urban grid. For better understanding of the turbulence recycling approach it would probably be helpful to indicate also the precursor grid.
- Author's changes in manuscript: Fig. 2 modified and a sentence referencing the figure is added to P7, l.11.
- (15) Referee comment: P6, 1.10 EC measurements IN Torni? The authors probably mean on top of the Torni building.

Author's response: Indeed.

Author's changes in manuscript: Corrected.

- (16) Referee comment: P6, l.13 by means of ...
- Author's changes in manuscript: Corrected.
- (17) Referee comment: P7, l.4 with a constant value

Author's changes in manuscript: Corrected.

- (18) Referee comment: P8, 1.10 the release of particles is activated: one would need to know how many particles are released (per time step, say and (probably) per grid cell.
- Author's response: The information about the particle release schedule is provided a bit later in Section 2.2.4 (LES-LS analysis).

Author's changes in manuscript: No changes.

- (19) Referee comment: P9, l.5 I don't think 'fixating on the exact location. . .' is very clear please reformulate
- Author's response: Perhaps 'fixating on' doesn't reflect the intended meaning of 'strictly/rigidly concentrating on' or 'strictly/rigidly focusing on'.

Author's changes in manuscript: Changed 'fixating' to 'strictly concentrating'.

(20) Referee comment: P10, l.2 reference simulation is run for 1 h 'to develop ABL turbulence sufficiently' (what is sufficiently?) and averaging is performed for the last 45 minutes. In other words, the effective spin-up time is 15 minutes. How does this compare to some eddy turnover time for the given situation? Can the authors comment?

- Author's response: This is incorrectly documented by the authors (see the first major comment) and we are pleased that this mistake was caught! The temporal averaging for extracting the mean profiles for the main run is optional and should *only* be used with precursor simulations that reach quasi-steady conditions. This is not the case in the precursor simulation performed for this application. The mean profiles used in the urban simulation are obtained from the flow field solution obtained after 1.5h (wrongly reported value here too!) of simulation time. Horizontally averaged profiles were monitored to determine when the desired boundary layer height and geostrofic wind value at  $z = \delta$  (P7. 1.5-6) were attained.
- Author's changes in manuscript: The appropriate corrections are made on P10, l.1-2. and on P7, l.6. and l.20.
- (21) Referee comment: P10, l. 14 to what do the computational costs amount in absolute terms?
- Author's changes in manuscript: The absolute measures in clock time and cpu hours are added to P10, 1.15.
- (22) Referee comment: P11, l.2 'each lth particle's coordinate. . ..': as this is formulated, not every particle is sampled (conditional on its position within the box) but only every lth particle. Is this what the authors want to say? And if so, what is 'l' and how is it determined? What is the reasoning not to sample all particles but a subset equally spaced in l?
- Author's response: Superscript l is declared as the identifier for the particles on P8, l.14. On P11, l.2 the intention is to say that "each particle's, labeled l, coordinate …" but the wording becomes cumbersome. Since the use of l in this context is redundant, we can remove it and just say "each particle's …".

Author's changes in manuscript:  $l^{\text{th}}$  removed from P11, l.2.

(23) Referee comment: P12, l.3 how large is  $\Delta x_f$  chosen?

Author's changes in manuscript:  $\Delta x_f = \Delta y_f = 2 \text{ m}$  added to P12, l.3.

(24) Referee comment: P13, l.1 close to the top

Author's changes in manuscript: Corrected.

- (25) Referee comment: P13, l. 12 what is 'negative far field'? Negative vertical velocity in the far field? Or negative footprint in the far (upwind) field)? Please specify.
- Author's response: The part containing this statement has now been rewritten due to the new results obtained while addressing one of the major comments.

Author's changes in manuscript: Part is already rewritten.

(26) Referee comment: P15, 1.10 the criterion

Author's response:

Author's changes in manuscript: Changed.

- (27) Referee comment: P16, 1.25 which is not sufficient...: based on what?
- Author's response: On P16, l.9-10 it is stated

"This reference footprint, labelled  $f^{\text{REF}}$ , should be constructed from at least  $10^6$  particle entries to facilitate a sufficiently informative evaluation of sensitivities."

Author's changes in manuscript: The sentence on P16, l.25 is modified for clarity.

(28) Referee comment: Fig 7, caption: incomplete (caption must include all information to understand the figure without reading the text).

Author's changes in manuscript: The caption is rewritten.

- (29) Referee comment: P19, l. 30 ... which would otherwise be employed: how can the authors state this? There are many different models of this kind, so this is only one of the models that possibly might be used.
- Author's response: The sentence was not supposed to highlight the particular analytical model used. A reformulation is certainly needed.
- Author's changes in manuscript: "an analytical model, which would otherwise be employed" changed to "an analytical model, which belongs to the group of closed-form models that would otherwise be employed"
- (30) Referee comment: Fig.9 caption: I don't think I can see any white circles. . ..
- Author's response: This is a leftover from an older version.

Author's changes in manuscript: Caption corrected.

(31) Referee comment: P22, l.1 that leads to

Author's changes in manuscript: Change applied.

(32) Referee comment: Fig. 11 the entries (labels) 'vegitation' (high and low) probably mean to refer to 'vegetation' and should be changed.

Author's changes in manuscript: Changes applied to the figure.

**References**

- A. Hellsten, S. Luukkonen, G. Steinfeld, F. Kanani, T. Markkanen, L. Järvi, T. Vesala, and S. Raasch. Footprint evaluation for flux and concentration measurements for an urban-like canopy with coupled lagrangian stochastic and large-eddy simulation models. *Bound-Lay. Meteorol.*, 157:191–217, July 2015.
- M. Letzel, C. Helmke, E. Ng, X. An, A. Lai, and S. Raasch. Les case study on pedestrian level ventilation in two neighbourhoods in hong kong. *Meteorologische Zeitschrift*, 21(6):575–589, 2012.
- A. Nordbo, L. Järvi, S. Haapanala, J. Moilanen, and T. Vesala. Intra-city variation in urban morphology and turbulence structure in Helsinki, Finland. *Boundary-Layer Meteorology*, 146:469–496, 2013.

**Response to Referee 2.**

Mikko Auvinen, Leena Järvi, Ullar Rannik, Antti Hellsten, Timo Vesala

June 26, 2017

**Referee #2 comments**

All page and line numbers refer to the *original* manuscript.

(1) Referee comment: How well can one determine the  $\bar{w}$  at the target volume and the uncertainty associated with it since this affects the uncertainty in the footprint through Eqs. (4) and (5). The authors argue that a well-defined value of  $\bar{w}$  cannot be determined at the target site for reasons discussed on pages 9 (lines 10 - 21) and 13 (lines 24 - 33), and that this limitation leads to the negative bias in the far-field footprint (Fig. 5). They go on to develop the far-field correction approach (pages 13 to 15), which involves a correction coefficient  $c_{ijk}$  applied to the original  $\bar{w}$ .

I believe that the authors should provide further information to quantify the uncertainty in  $\bar{w}$  at the target site (volume) and also upstream of it. In particular, they should compute and provide:  $\bar{w}, \sigma_w$ , and the uncertainty in  $\bar{w}$  (in a statistical sense) from their numerical wind field data: 1) over the urban landscape upstream of the target site but at the same height (or over the same heights) as the target volume [an areal average over the upstream domain or volume using a few heights to match those of the target volume and extending from the leading edge of the city,  $x \approx 2$  km (Fig. 2), to the Hotel Torni], and 2) over the target volume. The difference in the raw w values over the target volume (larger) and upstream area (smaller) presumably should lead to a larger  $\bar{w}$  and greater variance ( $\sigma_w^2$ ) over the target volume, fewer number of numerical data points, and greater uncertainty in  $\bar{w}$  over the target volume (e.g., uncertainty estimated as some factor  $\lambda$  depending on the w PDF times  $\sigma_w / \sqrt{N}$ , N being the number of points).

Furthermore, it would be useful to compute the subgrid-scale (sgs) turbulent kinetic energy (TKE,  $E_s$ ) and the LES resolved scale TKE ( $E_r$ ), and their ratio or an sgs rms velocity,  $\sigma_{sgs} = (2/3)E_s$  and the ratio,  $\sigma_{sgs}/\sigma_{wr}$ . This ratio would quantify the importance or not of the sgs velocities in determining the footprint; as with the  $\bar{w}$ ,  $\sigma_w$ , etc., this should be done for both the upstream area/volume and the target volume to make the distinction if there is one.

Author's response: This matter, concerning the problematic coordinate rotation applied to finite subvolumes, became resolved while investigating the Referee #1 comment (4). Please, refer to the associated author's reply.

The questions raised here concerning the uncertainty in  $\bar{w}$  were carefully considered in the preparation of the LES simulations. However, to reduce the amount of data output, only limited time-resolved datasets were written out while the 45 min time-averaged datasets using in the footprint computations were generated 'on the fly' within PALM during the simulation without specifically storing the individual snapshots. On these mean flow datasets statistics are not available. But, for visualization purposes we have stored two partially coinciding datasets that can be used to estimate some of the desired uncertainties. The first dataset (D1) is a  $(10 \text{ m})^3$  cube discretized at 1 m resolution centered above Hotel Torni (in a similar manner as the target box) containing velocity data gathered over 15 min at 1 Hz. See Fig. 1. The second dataset (D2) is a large *xz*-plane cutting across the center of the entire LES domain and through the Hotel Torni building containing velocity data over 45 min. However, this dataset is rather coarse for its spatial resolution is 4 m and the sampling rate 0.25 Hz. Figure 2 below illustrates a relevant part of this *xz*-plane and exemplifies a line along which velocity related statistics are gathered.

Within the D1 cube the vertical velocity data ranges [min, max] are  $\bar{w} = [-0.27, 4.50] \text{ m s}^{-1}$  and  $\sigma_w = [0, 1.22] \text{ m s}^{-1}$  whereas for the sample averages (averaged over the cube's volume) become  $\langle \bar{w} \rangle = 1.014 \text{ m s}^{-1}, \langle \sigma_w \rangle = 0.589 \text{ m s}^{-1}$ . We can estimate the standard deviation of the sample mean

Figure 1: Visualization of  $(10 \text{ m})^3$  cube discretized at 1 m resolution centered above Hotel Torni. This domain has been used to gather time accurate flow data for visualizations. Here it is used to provide estimates for quantifying the uncertainty in vertical velocity data. The figure depicts a slice through the dataset coloured by  $\sigma_w$ .

(or sampling error) for the 45 min (4 Hz) mean velocity dataset utilized in the footprint computation by using the  $\langle \sigma_w \rangle$  values from D1 but specifying  $N = 45 \times 60 \times 4 = 10800$  according to the 45 min logging schedule. This yields  $\sigma_w/\sqrt{N} = [0, 0.012] \,\mathrm{m \, s^{-1}}$  and  $\langle \sigma_w/\sqrt{N} \rangle = 0.0056 \,\mathrm{m \, s^{-1}}$ . Because the uncertainty in  $\bar{w}$  values used in the footprint computations is so small compared to the changes resulting from taking spatially averaged means  $\langle \bar{w} \rangle_{i,i,k}$  over subvolumes larger than one LES grid cell or applying the far field correction (where  $|c_{i,j,k} \langle \bar{w} \rangle_{i,j,k} - \langle \bar{w} \rangle_{i,j,k} | \approx 0.1 \,\mathrm{m \, s^{-1}}$ ), the uncertainty aspect did not enter into the method description. According to Wilczak et al. (2001) a typical sampling error values for a 15-min data runs are ca. 0.01-0.06 which can produce artificial tilt 'corrections' on the order of 0.5 degree. Since our sampling error for the 45 min data is roughly an order of magnitude smaller, the artificial tilt effect is deemed insignificant even when the piecewise post-processing is carried out in accordance with the LES grid resolution. The distributions of mean values and standard deviations obtained from D2 over a 1200 m line upstream of Hotel Torni's EC sensor are shown in Fig. 2 (at the end of the document), but this data is considered most relevant in the context of the last Referee comment. Now that the mechanism for causing the bias in the piecewise processed footprints has been established, and also for our inability to repeat this extremely expensive simulation to collect the requested information we do not have (e.g. subgrid scale and resolved scale TKE and upwind data), we wish to submit that the method description could be published without presenting the requested analysis.

- Author's changes in manuscript: The underlying issue here is addressed by the changes associated with the Referee #1 comment (4).
- (2) Referee comment: Section 2.3.2. This section is one of the more limiting parts of the paper in being somewhat arbitrary and subjective for generating the final LES-LS footprint. I believe that it is sufficient as an initial approach/procedure. However, for use by other researchers or as a standalone method for generating footprints (by others) for multiple EC sites in a city, a more rigorous, robust, and less subjective approach is necessary.

For example, on page 16, is the  $f^{(0)}$  with  $4 \times 10^5$  particle entries deemed insufficient as a reference footprint,  $f^{REF}$ , because there is too much blue (negative footprint value) in the far field of the footprint (Fig. 6)?

Conversely, is  $f^{(1)}$  judged to be sufficient and act as the reference because there is less blue? There should be some quantitative method, index, or variable that stipulates the adequacy of the reference (and why); e.g., requiring the far-field negative fooprint to be less than some small fraction of the maximum footprint. Such a criteria is used in deciding whether a candidate footprint,  $f^{(k)}$ , should be included in the final assembly (page 16, lines 31 - 33) with the  $||\Delta f||_{max}$  "determined according to the case-specific requirements." These requirements should be spelled out.

However, the maximum values,  $f_{max}^{(k)}$ , associated with these difference maxima,  $\delta f_{max}$ , and the fraction,  $\delta f_{max}/f_{max}^{(k)}$ , should be given. The  $||\Delta f||_{max}$  here was chosen from the  $f^{(3)}$  value, but it is not clear why. Also, the maximum footprint values in Fig. 6 appear to be about  $10^{-6}$  (bright red), but the  $||\Delta f||_{max} = 6.24 \times 10^{-3}$  is much larger. I don't understand this.

Author's response: The authors agree that the presented piecewise post-processing method presents a compromize between computational cost and precision. The presented methodology provides a computationally feasible technique to generate complex urban footprints that includes a mechanism to control (or at least monitor) the amount of imprecision in the final result. If the allowed tolerances for such complex footprints are made really strict, under the presented LES-LS framework the prize must be paid by increasing the computational cost by reducing the target volume to a single grid cell. But then again we return back to the problem of concentrating on a particular point  $\mathbf{x}_{M}$  over a building, which cannot be described in high detail by the topography model or the LES grid to begin with. It is our understanding that, even with 1 m LES resolution, it is unreasonable to assign 'analytical' precision requirements for the urban footprint problem. That said, it becomes important that the method provides a flexible way to generate sufficiently converged footprint candidates which contain different levels of imprecision. This provides the user of these footprints a possibility to assess how sensitive the considered application and the deduced conclusions are to the level of inexactness associated with the method. Therefore, the statement "determined according to the case-specific requirments" is not intended to be vague, but genuinely intentional. We are unable to dictate what the accuracy requirements are in all the possibile footprint applications. Instead, we believe that it is important that the method for generating an arbitrarily complex footprints offers the possibility to examine the level of inaccuracy in a transparent manner. For example, a standalone method utilizing the piecewise post-processing approach and selective assembly should generate a variety of footprints at different levels of confinement around  $\mathbf{x}_{M}$  – which means varying the number of particle entries in the footprint and, thereby, the level of convergence. An explicit statement concerning the possibility to vary the tolerances in the selective assembly phase for the purpose of assessing the associated impact in the footprint application is added to the manuscript.

In response to the other questions: The requirement that  $f^{\text{REF}}$  should contain around 106 particle entries stems from Hellsten et al. (2015) where it was established that 106 is the minimal amount to obtain a resonable footprint in an urban-like situation. In the same article footprints containing  $4 \times 10^5$  particle entries were shown to exhibit only the near field sufficiently. The computation of relevant 'deltas' between  $f^{\text{REF}}$  and  $f^{(l)}$  necessitates that at least the  $f^{\text{REF}}$  features a strongly converged near field. If both  $f^{\text{REF}}$  and  $f^{(l)}$  would contain, for instance,  $\sim 10^5$  particle entries, the  $||\Delta f^{(l)}||_{2,\Omega_*}$  values would be contaminated by the noise in the distributions.

The red-white-blue colormap chosen to illustrate the  $f^{(l)}$  candidates was intentionally set to have a really small magnitude for max  $(10^{-6})$  and made symmetric to keep white color at zero. This was done in effort to provide a clearer view of the spatial extent of the distributions in hopes that the obtained  $||\Delta f^{(l)}||_{2,\Omega_{\star}}$  value would reflect the qualitative discrepancy. For instance, our objective is to convey that the spatial pattern of  $f^{(5)}$ , especially its near field, is also evidently different from  $f^{\text{REF}}$ . In addition, the colormap manages to reflect the convergence level of the far field with different shades of 'grey' as the noise reduces and the values approach zero (white).

We did experiment with other metrics to assess discrepancies between two footprint candidates, but utilizing local maximum differences  $\delta f_{max}$  proved not useful at all. The obtained urban footprints feature rapid local changes when going from street canyons to roof-tops and when the convergence is not strong, some individual cells in the footprint grid exhibit exceptionally large deltas even though the overall agreement may be good. Therefore we submit that the  $\delta f_{max}$  values do not provide any guidance in the selection.

Author's changes in manuscript: Clarification about the required number of particles and the level of convergence in the footprints according to Hellsten et al. (2015) is added to P12, l.17. An additional statement concerning the maximum allowable discrepancy in the selective assembly is

added on P16, 1.33 and a general observation concerning the rejected contributions is added to P17, 1.4.

(3) Referee comment: page 2, line 4. "... which relates the value of a measurement (of flux or concentration) at location ..." The parenthetical words would make this clearer.

Author's response: Yes, agreed.

Author's changes in manuscript: Change applied to P2, 1.4.

- (4) Referee comment: page 2, Eq. (1). It should be made clear here that the footprint,  $f(x_M, x')$ , has the dimensions inverse of the integration unit(s). In Eq. (1), this is just length (L) since the integration is along a line in the domain, e.g., f could be the crosswind-integrated footprint,  $f^y$ , used later. But often, the footprint pertains to an elemental area dxdy ( $L^2$ ) as in Eq. (4).
- Author's response: In Eq. (1) the spatial variable is a vector (bold font type,  $\mathbf{x}'$ ) and the domain  $\Omega$  is a volumetric domain, whose vertical dimension is collapsed in the presentation. Thus, as the referee correctly stated, here the footprint has the dimensions inverse of the integration unit, thus  $m^{-3}$  and  $m^{-2}$  after collapsing  $\Omega$  into a sheet. Indeed, this should be clarified in the text.

**Author's changes in manuscript: Clarification added to P2. 1.7.**

- (5) Referee comment: page 3, lines 8, 9. "...conduct tracer gas experiments, which are nearly impossible to arrange in residential areas." However, this computational framework could be tested in some way in a large wind tunnel such as WOTAM at the University of Hamburg (e.g., Leitl and Schatzmann, 2010). This tunnel with dimensions of 4m × 3.2m × 25m (width, height, length) has been used to study many aspects of dispersion in large European cities. I would recommend that testing of the LES-LS numerical Helsinki footprint model be tested in some way in such a tunnel. It would give greater credibility to the approach especially when using the approach to suggest/demonstrate limitations in analytical footprint models.
- Author's response: We agree completely with the Referee that validation studies are needed for urban LES. Currently we are adopting the best practizes from Letzel (e.g., 2007); Letzel et al. (e.g., 2008, 2012); Hellsten et al. (e.g., 2015) while paying a very heavy prize for establishing the best possible inlet boundary conditions and resolving everything at 1 m resolution. However, we are in the process of completing an urban dispersion validation study at WOTAN wind tunnel featuring Hamburg site and are looking for other validation studies where more detailed wind data would be available.

**Author's changes in manuscript: We propose no changes.**

- (6) Referee comment: page 11, line 11. The authors talk generally about the target subvolumes, size, and number, but this should all be guided the size/scale of the turbulent flow structures at the EC sensor site. This was mentioned earlier in the paper, but no specifics were given. For example, on page 9 (line 32), it is merely stated that "the target box reasonably represents the sensor site." Should the target box scale or total volume be of the order of or less than (say some fraction of) the characteristic flow structure dimension/volume at the EC site? If we assume very crudely that the characteristic length scale is  $l_c = \kappa z$ , with  $\kappa$  being the von Karman constant (0.4) and z is height,  $l_c = 24$  m; i.e., for a neutral boundary layer over homogeneous terrain. The characteristic dimension of the sampling volume is  $l_{sc} = (l_{sx} \times l_{sy} \times l_{sz})^{1/3} = 12.4$  m, which is half of  $l_c$ . I don't know the logic of choosing the  $l_{sc}$  in the paper, but requiring it's overall dimension to be less or much less thant  $l_c$  may be a useful criterion. Clearly, there may be others.
- Author's response: We believe that this matter has now been clarified. Please, see author's response to Referee #1 comment (4). The turbulent length scales at the EC site strictly dictate the appropriate subvolume size requiring that, when no far field correction is employed, the subvolume size must adhere to the LES grid spacing. We did observe that further away from the sensor location (i.e. at the edges of the target volume where the mean flow gradients have subdued) the resolution requirement could be relaxed from 1 m to 2 m.
- Author's changes in manuscript: Modified the statement on P11, l.11 to emphasize that the complexity of the flow solution in the vicinity of  $\mathbf{x}_{M}$  dictates the resolution of the subvolumes. Also, the rewriting of Section 2.3.1 addresses this issue as well.

- (7) Referee comment: page 12, line 19. "distributions" Do you mean individual "footprint" distributions?
- Author's response: Yes, the original statement refers to individual footprint distributions. This part has now also been rewritten as a result of the responses to Referee #1.
- Author's changes in manuscript: This statement has been replaced.
- (8) Referee comment: page 14, line 3. "negligibly small offset." But that offset is 20% to 25% of the maximum  $\bar{f}^y$  in Fig. 5, and thus is not small.
- Author's response: Referring to the sentence "If the footprint distribution plateaus in the far field, this asymptote can be declared as the zero reference level, which deviates from the 'correct' asymptote by a negligibly small offset" the intended meaning is that if a footprint has a far field that has plateaued, but with a large offset (for example 20% to 25%), it is not physically justifiable. Only a near zero far field asymptote is acceptable. Thus, we should impose that the plateaued far field can only occur if it approaches zero. Now, in reality, the footprint asymptote is not exactly zero and therefore a zero-imposed far field deviates from the near-zero far field by a small amount.
- Author's changes in manuscript: In hopes to clarify the statement, we changed "can be declared as" into "can be amended to become".
- (9) Referee comment: page 15, Eq. (9). It might be noted that a similar approach is often used in determining the vertical tilt axis of a sonic anemometer on a meteorological tower; i.e., due to the mounting imprecision and/or possible variability in the sensor vertical axis over time (wind loads, vibrations, etc). Assuming that over a long time record above chosen flat terrain the  $\langle \bar{w} \rangle = 0$ , the tilt axis (**c** or  $c_{ijk}$ ) (relative vertical velocity to the 3 components fixed coordinates) is chosen by ensuring  $\sum_{n}^{N} \mathbf{c} \cdot \mathbf{u}_{n} = 0$  (sum of the vertical velocity components is zero), where *n* is the measurement (realization or time) index and *N* is the total number of measurements.
- Author's response: Yes, the resemblance is notable. Perhaps not immediately apparent, but worth mentioning.

Author's changes in manuscript: A sentence added to P15, l.12.

- (10) Referee comment: page 20, lines 8 29, Fig. 9. Why is the numerical footprint,  $f^y$ , so much more peaked than the analytical Korman and Meixner (KM) (2001) model? There are at least two potential reasons for this as suggested by the authors. 1) The real urban terrain/fetch upstream of Hotel Torni is only about 1600 m or about 39% of the total LES domain (4096 m; Fig. 2). The real terrain which is heterogeneous with a marked change in roughness at the upwind edge of the city is accounted for in the LES, whereas the KM presumably assumes homogeneous conditions upstream. These very different mean wind and turbulence fields could possibly be accounted for in the KM model in a crude way (point 9 below). 2) The mean wind and rms lateral turbulence component ( $\sigma_v$ ) for the KM model were extracted from the measurements at the top of the Hotel Torni. The authors should repeat the KM calculation using the LES mean wind and  $\sigma_v$  at the height of Hotel Torni. This could be done using both the LES values 1) within the target volume and 2) the upwind U and  $\sigma_v$  values. That is, there should be consistency in the meteorological inputs to the LES-LS and the KM model.
- Author's response: Initially the KM model input parameters were taken from the LES solution, but this was changed because we felt that it would be undefendable to utilize data that would not be available under normal circumstances when analytical models would be used. Fortunately, the tower and LES data agrees to a remarkable degree. If we collect LES data from a small cube, say  $(2 \text{ m})^3$ , or a line around  $\mathbf{x}_{\text{M}}$  and take the sample average, we get  $\sigma_v = 0.72 - 0.74 \text{ m s}^{-1}$  and  $U = 4.5 - 5.1 \text{ m s}^{-1}$  (U changes rapidly over the tower) depending on the method. Comparing these to the tower data  $\sigma_v = 0.75 \text{ m s}^{-1}$  and  $U = 4.86 \text{ ms}^{-1}$  while considering that the LES simulation gets its meteorological boundary conditions from Lidar (which only provided us the geostrofic wind and the height of the boundary layer) one gets easily excited. (It's a pity we only have one 'tower' in the simulation domain.)

However, as the referee points out, the  $\sigma_v$  and U values could also be taken from the LES flow field upstream of Hotel Torni. This would certainly provide a more representative description of the flow conditions above the effective source area. But is this a justifiable approach when the objective is to assess "the potential error that may arise when analytical, closed-form footprint models are applied to urban flux measurements"? One would need to perform high resolution LES analysis (as documented here) or have access to multiple tower measurements in the urban canopy to acquire this information. This brings us to a very relevant topic: How to best utilize analytical footprint models in urban measurement campaings? As it is in everyone's best interest (including our's) to 'save' the applicability of analytical models to urban measurements, these considerations naturally will follow and LES-based methods will play a critical role in these investigations. Nevertheless, this publication cannot extend its scope to this important and difficult topic. Devising a technique that allows a real urban footprint to be generated, alone, proved to be a substantial task.

But, since we do have a coarsely resolved (4 m resolution and 0.25 Hz) dataset of a large xz-plane cutting across Hotel Torni, see Fig. 2 below, we can demonstrate the difference that results when upwind values are employed instead. The dataset was intended for visualization purposes, but offers reasonable accuracy for such a large scale investigation. We chose a 1200 m long line (also visible in Fig. 2), which approximately stays 60 m above the ground level, which rises 14 m from the sea shore to Hotel Torni. Computing the line averages gave for  $\bar{u} = 6.4 \,\mathrm{m \, s^{-1}}$  (i.e. U) and  $\sigma_v = 0.95 \,\mathrm{m \, s^{-1}}$ . A juxtaposition of the different KM footprint distributions are shown below in Fig. 3. The effect is visible (with carefully selected colormap) but in the source area fractions the difference amount to only 0.1% in building, road and impervious land cover types while the rest remained unchanged. In the light of this evidence, we submit that these results are not included in this publication.

---

## Author Response (AR2)

**Response to Referee's Minor Comments**

Mikko Auvinen, Leena Järvi, Üllar Rannik, Antti Hellsten, Timo Vesala

September 14, 2017

**Minor comments**

The page and line numbers within square brackets [] refer to the *diff* document (used by the Referee) and the ones marked within parenthesis () correspond to the *Rev. 1* of the manuscript.

**(1) Referee comment:** [P9, l.33] (P9, l.33) to represent

**Author's changes in manuscript:** Corrected.

**(2) Referee comment:** [P11,l.9] (P11, l.7) due to my comment the authors have delete 'lth' but this makes the following notation quite unclear. I suggest: 'each particle's (identified as '*l*') coordinate ...'

**Author's changes in manuscript:** Statement modified according to the suggestion, but without the apostrophies around $l$.

**(3) Referee comment:** [P11, l.30] (P11, l.28) particle entries

**Author's changes in manuscript:** Corrected.

**(4) Referee comment:** [P12, l.1] (P11, l.30) of particle l

**Author's changes in manuscript:** "... deviation of the $l^{\text{th}}$ particle from ..." changed to "... deviation of particle $l$ from ...".

**(5) Referee comment:** [P14, l.31] (P13,l.28) the individual sectional sectional (better: sectoral) footprints inadequately converge and thereby become

**Author's response:** The word 'sectional' carries the intended meaning here and therefore we modify the sentence using that word.

**Author's changes in manuscript:** Sentence on (P13,l.28) modified according to the comment.

**(6) Referee comment:** [P16, l.3] (P14, l.13) is expected to fall ...

**Author's changes in manuscript:** Corrected.

**(7) Referee comment:** [P17, l.3] (P15, l.21) give rise to an error

**Author's changes in manuscript:** Corrected.

**(8) Referee comment:** [P18, l.34] (P17, l.26) such a result

**Author's changes in manuscript:** Corrected.

**(9) Referee comment:** Tab 3 measurement height: 55.1 m. However on [P3, l.19] (P3, l.19): measurement height is 60 m agl and on [P22, l.13] (P21, l.7): displacement height is 14.9 m: this yields a measurement height of 45.1 m. First this needs to be made consistent and second, possibly, the KM simulation (in Fig. 8) needs to be redone.

**Author's response:** The measurement height value in Tab 3 is a typo, whereas the computations of the KM model were carried out with correct values (which are actually hard-coded in the released version of the P4UL python library and its *footprintGather.py* script):

```
L =10000.; z_m = (60.-14.9); z_0 = 1.4; sigma_v = 0.75; u=4.86
x_off = 2.*228.; y_off = 2.*508.
F_km  = kormann_and_meixner_fpr(z_0, z_m, u, sigma_v, L, Xt, Yt, x_off, y_off)
```

**Author's changes in manuscript:** Correct value (45.1 m) is added to Tab 3. Also, the wording used on [P3, l.19-20] is improved for better legibility.

**(10) Referee comment:** [P26, l.6] (P25, l.7) depends on

**Author's changes in manuscript:** Corrected.

**(11) Referee comment:** [P26, l.21] (P25,l.22) leads to prohibitive computational costs

**Author's changes in manuscript:** Corrected.

[revised manuscript text omitted]